# Learning Graph Invariance by Harnessing Spuriosity

**Tianjun Yao**[1] **Yongqiang Chen**[1,2] **Kai Hu**[2] **Tongliang Liu**[3,1] **Kun Zhang**[1,2] **Zhiqiang Shen**[1]

[1]Mohamed bin Zayed University of Artificial Intelligence
[2]Carnegie Mellon University
[3]The University of Sydney
{tianjun.yao, yongqiang.chen}@mbzuai.ac.ae, kaihu@cs.cmu.edu
tongliang.liu@sydney.edu.au, {kun.zhang, zhiqiang.shen}@mbzuai.ac.ae

## Abstract

Recently, graph invariant learning has become the *de facto* approach to tackle the Out-of-Distribution (OOD) generalization failure in graph representation learning. They generally follow the framework of invariant risk minimization to capture the invariance of graph data from different environments. Despite some success, it remains unclear to what extent existing approaches have captured invariant features for OOD generalization on graphs. In this work, we find that representative OOD methods such as IRM and VRex, and their variants on graph invariant learning may have captured a limited set of invariant features. To tackle this challenge, we propose **LIRS**, a novel learning framework designed to **L**earn graph **I**nvariance by **R**emoving **S**purious features. Different from most existing approaches that *directly* learn the invariant features, LIRS takes an *indirect* approach by first learning the spurious features and then removing them from the ERM-learned features. We demonstrate that learning the invariant graph features in an *indirect* way enables the model to capture a more comprehensive set of invariant features, leading to better OOD generalization performance in novel environments. Notably, LIRS surpasses the second-best method by as much as $25.50\%$ across all competitive baselines, underscoring its efficacy in OOD generalization. [1]

## 1 Introduction

Graph representation learning with graph neural networks (GNNs) (Kipf & Welling, 2017; Xu et al., 2019; Veličković et al., 2017) has proven to be highly successful in tasks involving relational information (Qiu et al., 2018; Wu et al., 2022b; Yu et al., 2018; Zhang et al., 2022c). Despite their achievements, GNNs often assume that the training and test data share the same distribution. This assumption often does not hold in real-world applications (Hu et al., 2020; Huang et al., 2021; Ji et al., 2022; Koh et al., 2021) due to changes in the underlying data generation process, leading to distribution shifts. Such phenomenon can severely degrade the performance of GNN models, which presents a critical challenge for their deployment in practical scenarios (DeGrave et al., 2020).

To address the OOD generalization challenge, graph invariant learning has become the *de facto* approach. Inspired by the success of invariance principle (Arjovsky et al., 2020; Kreuzer et al., 2021), graph invariant learning aims to identify invariant subgraphs that are causally related with the targets (Yang et al., 2022; Li et al., 2022b; Liu et al., 2022; Zhuang et al., 2023; Chen et al., 2022; Li et al., 2022a). Typically, they follow the framework of Invariant Risk Minimization (IRM) (Arjovsky et al., 2020) or Variance-Risk Minimization (VRex) (Krueger et al., 2021), which aim to capture invariant features by learning an equipredictive classifier across different environments. Despite some success, many graph invariant learning methods only perform similarly compared to the traditional Empirical Risk Minimization (ERM) (Vapnik, 1995) in graph OOD benchmarks (Gui et al., 2022), it raises a critical yet overlooked question:

*To what extent do the graph invariant learning algorithms capture the graph invariance?*

---

[1]Code is available at `https://github.com/tianyao-aka/LIRS-ICLR2025`

In this work, we address this question by investigating two representative OOD algorithms, namely IRMv1 (Arjovsky et al., 2020) and VRex (Kreuzer et al., 2021), in graph-level OOD classification scenarios. These algorithms are widely used, and many graph invariance learning algorithms are based on their variants or extensions (Li et al., 2022b; Liu et al., 2022; Zhuang et al., 2023; Wu et al., 2022c). Surprisingly, we find that IRM, VRex and their variants on graph invariant learning may only learn *a subset* of invariant features (Sec. 3). In contrast, the learning paradigm that learns invariant features *indirectly* by removing spurious features from the ERM-learned features may render learning a more comprehensive set of invariant features (Figure 1 and Sec. 3). Motivated by this observation, we propose a novel learning framework **LIRS**: **L**earning graph **I**nvariance by **R**emoving **S**puriosity. Different from most existing approaches that *directly* learn the invariant features, LIRS adopts the *indirect* learning paradigm. Specifically, LIRS first leverages the *biased infomax principle* (Def. 3) to effectively learn graph spuriosity (Theorem 4.1), then employees *class-conditioned cross entropy loss* to effectively learn graph invariance (Theorem 4.2). Extensive experiments on synthetic datasets demonstrate that LIRS is able to learn more invariant features compared to state-of-the-art graph invariant learning methods that adopt the direct invariant learning paradigm. Furthermore, LIRS shows superior OOD performance on real-world datasets with various types of distribution shifts, highlighting its effectiveness in learning graph invariant features. Our contributions can be summarized as follows:

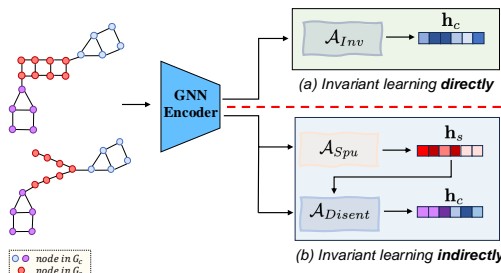

Figure 1: Illustration of the two graph invariant learning paradigms. Compared with the invariant learning methods $\mathcal{A}_{Inv}$ that adopt the direct learning paradigm, our proposed method first adopts a spuriosity learner $\mathcal{A}_{Spu}$ to learn spurious features, then learns invariant features by removing the spurious features using the feature disentanglement module $\mathcal{A}_{Disent}$.

- We reveal that learning invariant features *indirectly* i.e., by first learning and then removing spurious features, can be more effective in capturing invariant features than existing (graph) invariant learning algorithms that learn invariant features *directly*.

- We propose **LIRS**, a novel framework that adopts the indirect learning paradigm for learning invariant features, which consists of: a) The *biased infomax principle*, a contrastive learning algorithm that provably learns graph spuriosity; b) *class-conditioned cross-entropy loss*, a learning objective that effectively learns graph invariance by removing spurious features from ERM-learned features.

- Extensive experiments demonstrate that LIRS outperforms second-best baseline methods by up to 25.50% across 17 competitive baselines on both synthetic and real-world datasets with various distribution shifts.

## 2 PRELIMINARY

**Notations.** An undirected graph $G$ with $n$ nodes and $m$ edges is denoted by $G := \{\mathcal{V}, \mathcal{E}\}$, where $\mathcal{V}$ is the node set and $\mathcal{E}$ is the edge set. $G$ is also represented by the adjacency matrix $\mathbf{A}$ and the node feature matrix $\mathbf{X} \in \mathbb{R}^{n \times D}$ with $D$ feature dimensions. We use $G_c$ and $G_s$ to denote invariant and spurious subgraph for graph $G$ respectively, and $\mathbf{h}_c$ and $\mathbf{h}_s$ are invariant and spurious features in the latent space. $\widehat{\mathbf{h}}_i$ and $\widehat{\mathbf{h}}_G$ denote the estimated representations for node $v_i$ and graph $G$ respectively. Finally, we use $[K] := \{1, 2, \ldots, K\}$ to denote an index set, $w$ to denote a scalar value, $\mathbf{w}$ to denote a vector, $\mathbf{W}$ to denote a matrix, $W$ to denote a random variable, and $\mathcal{W}$ to denote a set. A more complete set of notations is presented in Appendix A.

**Problem Definition.** We focus on OOD generalization in graph classification in hidden environments. Given a set of graph datasets $\mathcal{D} = \{\mathcal{G}^e\}_{e \in \mathcal{E}_{tr} \subseteq \mathcal{E}_{all}}$, a GNN model $f$, denoted as $\rho \circ h$, comprises an encoder $h : \mathbb{G} \to \mathbb{R}^F$ that learns a representation $\mathbf{h}_G$ for each graph $G$, followed by a downstream classifier $\rho : \mathbb{R}^F \to \mathbb{Y}$ to predict the label $\widehat{Y}_G = \rho(\widehat{\mathbf{h}}_G)$. The objective of OOD generalization on graphs is to learn an optimal GNN model $f^*(\cdot) : \mathbb{G} \to \mathbb{Y}$ using data from training environments $\mathcal{D}_{tr} = \{\mathcal{G}^e\}_{e \in \mathcal{E}_{tr}}$ that effectively generalizes across all (unseen) environments:

$f^*(\cdot) = \arg\min_f \sup_{e \in \mathcal{E}_{\text{all}}} \mathcal{R}(f \mid e)$, where $\mathcal{R}(f \mid e) = \mathbb{E}_{G,Y}^e[l(f(G), Y)]$ is the risk of the predictor $f$ on the environment $e$, and $l(\cdot, \cdot) : \mathbb{Y} \times \mathbb{Y} \to \mathbb{R}_+$ denotes a loss function.

**Assumption 1.** (**Stable and predictive subgraphs**) Given a graph $G \in \mathcal{D}$, there exists a set of stable (causal) substructure patterns $\mathcal{G}_c$ for every class label $Y = y$, satisfying: a) $\forall e, e' \in \mathcal{E}_{tr}, G_c \in \mathcal{G}_c, \mathbb{P}^e(Y \mid G_c) = \mathbb{P}^{e'}(Y \mid G_c)$; and b) The target $Y$ can be expressed as $Y = f^*(G_c) + \epsilon$, where $\epsilon \perp\!\!\!\perp G$ represents random noise, and $\perp\!\!\!\perp$ indicates statistical independence.

Assumption 1 posits that one or more substructure patterns in $\mathcal{G}_c$ are not only stably associated with the target label $Y$ across different environments but also possess sufficient predictive power to accurately determine $Y$. This assumption is well-aligned with real-world scenarios. For instance, in the GOODHIV dataset (Gui et al., 2022; Hu et al., 2020; Wu et al., 2018), a molecule's ability to inhibit HIV may depend on the presence of several functional groups interacting with various parts of the virus. Moreover, recent study also provides empirical evidence for graph applications, suggesting that multiple substructures remain stable and predictive of the targets (see Appendix E.2 in Bui et al. (2024)), thereby supporting the validity of Assumption 1. Therefore, when the OOD algorithms are able to learn a broader set of invariant substructures (features), they will generalize more effectively across different environments.

## 3 WHY LEARNING INVARIANT FEATURES INDIRECTLY?

In this section, we motivate our study by addressing the question: *Why might learning invariant features indirectly be more effective than learning them directly?* We begin by presenting an intuitive hypothesis related to our research question, followed by an empirical study on synthetic datasets to support this hypothesis. Consider a dataset $\mathcal{D}$ where each sample $G^{(i)}$ com-

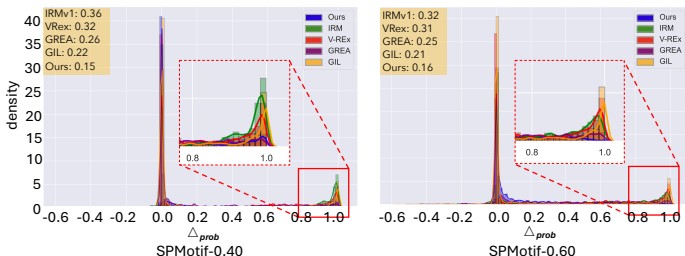

Figure 2: Comparison of the distribution $\mathbb{P}_{\Delta_{prob}}$ across different OOD algorithms.

prises invariant features $\{\mathbf{h}_{c,j}^{(i)}\}_{j=1}^{n_c}$ and spurious features $\{\mathbf{h}_{s,j}^{(i)}\}_{j=1}^{n_s}$, with $n_c$ and $n_s$ denoting the number of invariant and spurious features respectively. Any subset of the invariant features $\{\mathbf{h}_{c,j}^{(i)}\}_{j=1}^{n_c}$ maintains stable relationships with $Y$, consequently OOD objectives that aim to directly learn invariant features (e.g., IRMv1 and VREx) may only capture a subset of these invariant features if a subset of invariant features can already achieve accurate predictions in the training set. In contrast, an OOD objective that seeks to remove the spurious features $\{\mathbf{h}_{s,j}^{(i)}\}_{j=1}^{n_s}$ from the ERM-learned features may result in learning a more complete set of invariant features from $\{\mathbf{h}_{c,j}^{(i)}\}_{j=1}^{n_c}$, thus facilitating OOD generalization in unseen environments.

**Empirical Study.** To validate our hypothesis that learning invariant features by removing spurious features may facilitate learning more invariant features, we perform experiments using a variant of SPMotif datasets (Wu et al., 2022c; Ying et al., 2019). Specifically, for each sample three invariant subgraphs are attached to the base spurious subgraph in the training and validation sets. In the test dataset, we perform model inference and record the estimated probability for *correctly predicted samples*, denoted as $\hat{y}_j^{(i)}$. Subsequently, we randomly removed two invariant subgraphs from these samples and compute the new estimated probability score, denoted as $\tilde{y}_j^{(i)}$. We then fit the distribution $\mathbb{P}_{\Delta_{prob}}$ using Kernel Density Estimation (KDE) (Terrell & Scott, 1992), based on the changes in probability $\Delta_{prob} := \hat{y}_j^{(i)} - \tilde{y}_j^{(i)}$, $\forall i, j$, as illustrated in Figure 2. If the encoder $h(\cdot)$ can indeed learn all invariant features in the test set, then removing two invariant subgraphs should not change $\hat{y}_j^{(i)}$. Therefore, the closer $\mathbb{P}_{\Delta_{prob}}$ is to 0, the more it indicates that the encoder has learned the full set of invariant features. In addition to the density function of $\Delta_{prob}$, we also calculate the Wasserstein distance of each algorithm to a null distribution, where the probability mass is entirely centered at 0. As demonstrated in Figure 2, IRMv1 (Arjovsky et al., 2020) and VRex (Krueger et al., 2021) exhibit significant changes in $\Delta_{prob}$, implying that for some samples, IRMv1 and VRex learn

only a subset of the invariant patterns. GIL (Li et al., 2022b) and GREA (Liu et al., 2022) are two variants of VRex and IRM that learn invariant features directly by performing environment inference and environment augmentation respectively. As shown in Figure 2, the Wasserstein distances for GIL (Li et al., 2022b) and GREA (Liu et al., 2022) are also greater than that of our proposed method. This highlights the advantages of the learning paradigm that indirectly learns invariant features by first learning and then removing spuriosity. However, one may raise concerns that while

learning invariant features by first removing spurious features may indeed lead to capturing a more complete set of invariant features for the correctly predicted samples, it may also introduce spurious patterns if the spurious features are not effectively removed, which could render such a learning paradigm suboptimal in practice. To address this concern, we report test accuracy under varying strengths of spurious correlations, following previous experiments. As the strength of the spurious correlation increases, it becomes more challenging to eliminate the spurious features. From Figure 3, we have three observations: a) Our method consistently achieves superior OOD performance across different correlation strengths; b) Our method is less sensitive to the increased spurious correlations compared to other representative OOD methods, which is evident by the flatter curve in Figure 3; c) The higher test accuracy, combined with the distribution of

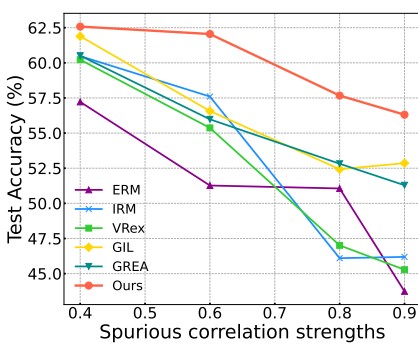

Figure 3: Test accuracy under varying spurious correlation strengths of the variant of SPMtoif datasets.

probability changes from the previous experiment, suggests that our method enables a larger number of test samples to learn a more complete set of invariant features. Based on these observations and analysis, we conclude that indirectly learning invariant features is more effective than learning them directly.

## 4 METHODOLOGY

Motivated by the observations in Sec. 3, we now introduce our proposed **LIRS** framework, which learns invariant features on graphs *indirectly*.

### 4.1 LEARNING GRAPH SPURIOSITY VIA BIASED INFOMAX

The first step of learning invariant features in the LIRS framework is to effectively learn graph spuriosity. In this section, we first propose a variant of the infomax principle (Hjelm et al., 2019; Veličković et al., 2019; Linsker, 1988), namely *biased infomax*, which encourages spuriosity learning by suppressing invariant signals. We then present a more practical version of biased infomax, suitable for the real-world datasets. We first formally define *spuriosity learning* in below:

**Definition 1.** (*Spuriosity Learning*) Assuming each data instance $G^{(i)}$ in a dataset $\mathcal{D}$ is composed of two parts $\mathbf{h}_c^{(i)}$ and $\mathbf{h}_s^{(i)}$, an algorithm $\mathcal{A}$ is said to be a spuriosity learner if $\mathbf{h}_s^{(i)} = \mathcal{A}(G^{(i)})$, $\forall i$, i.e., $\mathcal{A}$ learns only the spurious features for all instances in $\mathcal{D}$.

One such algorithm to encourage spuriosity learning is the *global-local infomax principle* (Hjelm et al., 2019; Veličković et al., 2019; Linsker, 1988), which maximizes the mutual information (MI) between global representation and local node representations, as defined in the following:

**Definition 2.** (*The Infomax Principle*) The infomax principle optimizes the following optimization objective in Eqn. 1 w.r.t the GNN encoder $h_\theta(\cdot)$.

$$\max_\theta \ \mathbb{E}_{G \sim \mathcal{G}} \frac{1}{|G|} \sum_{v_i \in G} I\left(\widehat{\mathbf{h}}_i, \widehat{\mathbf{h}}_G\right), \ s.t. \ \widehat{\mathbf{h}}_i = h_\theta(G), \widehat{\mathbf{h}}_G = READOUT(\widehat{\mathbf{h}}_i). \tag{1}$$

The encoder $h_\theta(\cdot)$ will primarily learn spurious features (Yao et al., 2024). However, it will still be constrained by $|G_c|$, the size of the invariant subgraph. As $|G_c|$ increases, the learned global representation $\widehat{\mathbf{h}}_G$ will increasingly capture invariant representations (Yao et al., 2024). To mitigate this constraint and further facilitate spuriosity learning, we propose the *biased infomax principle*.

**Definition 3.** (*The Biased Infomax Principle*) The biased infomax principle optimizes the following optimization objective in Eqn. 2 w.r.t the GNN encoder $h_\theta(\cdot)$.

$$\max_\theta \ \mathbb{E}_{G\sim\mathcal{G}} \frac{1}{|G|} \left( \sum_{v_i \in G_s} I\left(\widehat{\mathbf{h}}_i, \widehat{\mathbf{h}}_G\right) - \sum_{v_i \in G_c} I\left(\widehat{\mathbf{h}}_i, \widehat{\mathbf{h}}_G\right) \right), \ \text{s.t. } \widehat{\mathbf{h}}_i = h_\theta(G), \ \widehat{\mathbf{h}}_G = \text{READOUT}(\widehat{\mathbf{h}}_i).$$

(2)

By maximizing MI across nodes $v_i \in G_s$ and minimizing MI across nodes $v_i \in G_c$, the biased infomax principle yields an encoder $h_{\theta^*}(\cdot)$ that encourages the learning of spurious patterns while suppressing invariant ones. Formally, we can demonstrate that the biased infomax principle achieves spuriosity learning.

**Theorem 4.1.** *Given that the invariant subgraph $G_c$ contains invariant patterns that are causally related to the target labels, and $G_s$ contains only spurious patterns, the biased infomax principle achieves spuriosity learning, i.e., the encoder $h_{\theta^*}(\cdot)$ learns solely the spurious features for each data sample $G$ in $\mathcal{D}$.*

To achieve invariant learning by removing spuriosity, it is crucial to effectively learn the spurious features first. Theorem 4.1 demonstrates that the biased infomax principle can learn graph spuriosity more effectively the vanilla infomax, as defined in Def. 2. However, in real-world scenarios, $G_c$ is often unobservable, which hinders the practicality of the biased infomax principle. To make the proposed algorithm more practical, we propose an instance-level adaptive biased infomax, which incorporates a GNN explainer to identify the important nodes first, and then performs instance-wise counterfactual inference to determine whether the selected nodes should be biased or not.

**Instance-level adaptive biased infomax.** At first glance, the biased infomax principle appears impractical due to the necessity of knowing $G_c$ and $G_s$. To address this limitation, we employ an approximation algorithm to identify $\widehat{G}_c$, which serves as an estimate for $G_c$. Subsequently, we treat nodes $v_i \in \widehat{G}_c$ to realize the biased infomax principle. In practice, we utilize GSAT (Miao et al., 2022), a GNN explainer $e(\cdot)$ that is robust under distribution shifts, to identify $\widehat{G}_c$ and $\widehat{G}_s = G \setminus \widehat{G}_c$. Nevertheless, the approximation error may inadvertently bias the algorithm by amplifying invariant signals and suppressing spurious ones, as demonstrated in the following proposition.

**Proposition 1.** *Given an error rate $p\%$ in the approximation algorithm for $G_c$, i.e., $\widehat{G}_c$ contains $1 - p\%$ nodes from $G_c$ and $p\%$ nodes from $G_s$, let the learning objectives for the biased infomax with the ground-truth subgraphs $G_c$ and $G_s$ and with the approximated subgraphs $\widehat{G}_c$ and $\widehat{G}_s$ be denoted as $\mathcal{L}(\theta^*; \mathcal{D})$ and $\mathcal{L}(\theta'; \mathcal{D})$ respectively. The difference between $\mathcal{L}(\theta^*; \mathcal{D})$ and $\mathcal{L}(\theta'; \mathcal{D})$ can be expressed as:*

$$\mathbb{E}_{G\sim\mathcal{G}} \left[ \frac{2}{|G|} \left( \sum_{v_i \in pG_s} I\left(\widehat{\mathbf{h}}_i; \widehat{\mathbf{h}}_G\right) - \sum_{v_i \in pG_c} I\left(\widehat{\mathbf{h}}_i; \widehat{\mathbf{h}}_G\right) \right) \right].$$

Here $pG$ denotes $p\%$ nodes in graph $G$. When $\widehat{\mathbf{h}}_G$ learns primarily spurious features, $I\left(\widehat{\mathbf{h}}_i; \widehat{\mathbf{h}}_G\right) > I\left(\widehat{\mathbf{h}}_j; \widehat{\mathbf{h}}_G\right), \forall v_i \in G_s, v_j \in G_c$, then Prop. 1 implies that as $p\%$ increases, the gap between the ground-truth learning objective and the learning objective with approximation will also increase, thus leading to inaccurate graph representations. To reduce the approximation error and mitigate this issue, we conduct *counterfactual inference* for a graph $G$ by evaluating the changes in predicted probability with and without $\widehat{G}_c$ derived from the GNN explainer. Assuming $\widehat{G}_c = e(G, Y)$ closely approximates the ground-truth $G_c$, the removal of $\widehat{G}_c$ should result in a significant change in the predicted probability score. Under this condition, for graph $G$, we bias $v_i \in \widehat{G}_c$. Otherwise, we employ vanilla infomax without the bias operation. Specifically, we use a pre-defined threshold $\tau$. For each graph data instance $G^{(i)}$ with target label $j$, let $\widehat{y}_j^{(i)}$ and $\tilde{y}_j^{(i)}$ denote the predicted probability scores before and after the removal of $\widehat{G}_c$, respectively. We then bias the infomax for graph $G^{(i)}$ if $\widehat{y}_j^{(i)} - \tilde{y}_j^{(i)} > \tau$ using Eqn. 2; otherwise, we use Eqn. 1 to learn the graph representation.

## 4.2 LEARNING GRAPH INVARIANCE VIA CLASS-CONDITIONED CROSS-ENTROPY LOSS

**Class-conditioned cross-entropy loss.** So far we have obtained the spurious graph representations $\mathbf{h}_s$ for every instance $G$ in the dataset. Next we consider how to learn graph invariance by removing the spurious features from ERM-learned representations. To solve this challenge, previous study (Yao et al., 2024) takes spurious features $\mathbf{h}_s$ and target label $Y$ as inputs to train a linear classifier using cross-entropy loss to obtain the logits, which capture spurious correlation for each sample $G$, and use these (spurious) logits for subsequent spurious feature removing. Although the input only contains spurious features, we argue that: *The logits generated may be inaccurate to capture the spurious correlation strengths due to the overlapping of spurious patterns across different classes.* To see this, we propose Prop. 2 to demonstrate this issue:

**Proposition 2.** *Given a linear regression model with parameters $\{\theta_1, \theta_2\}$ and spurious features $\{x_1, x_2\}$, the correlation strength for feature $x_1$ is $p$ and for $x_2$ is $1 - p$ when $Y = 0$. Similarly, the correlation strength for feature $x_1$ is $q$ and for $x_2$ is $1 - q$ when $Y = 1$. Assuming the spurious features $x_1$ and $x_2$ can each take values in $\{0, 1\}$, we obtain the following parameter estimates using Mean Squared Error (MSE) loss: $\theta_1 = \frac{q}{p+q}, \theta_2 = \frac{1-q}{2-p-q}$.*

Prop. 2 demonstrates the issue brought by the interference of overlapping spurious features across different classes. For instance, when $p = q$, $\theta_1 = \theta_2 = \frac{1}{2}$, the generated logits fail to distinguish different spurious patterns. To mitigate this issue, we propose *class-conditioned cross-entropy loss*. Specifically, we first perform clustering within each class based on the spurious features $\{\mathbf{h}_s^{(i)}\}_{i=1}^n$. The resulted clusters will reflect different spurious patterns and environments accurately given the spurious features learned by the biased infomax objective. However, employing hard labels can result in a loss of information regarding the relative positions of samples within a cluster. To address this, we refit a linear classifier using the cluster labels as targets. This approach ensures that samples near the cluster boundaries exhibit higher entropy in their spurious logits, thereby preserving the information about their position within the cluster. Given the estimated spurious logits $\mathbf{s}^{(i)} \in \mathbb{R}^K$ with $K$ clusters in each class, we propose the following learning objective for spurious feature removing:

$$\mathcal{L}_{\text{inv}} := \mathbb{E}_{y \sim \mathcal{Y}} \, \mathbb{E}_{G^{(i)}|Y=y} \, -\sum_{j=1}^{K} w^{(i)} \mathbf{s}_j^{(i)} \log\left(\sigma(\widehat{\mathbf{s}}_j^{(i)})\right), \, s.t., \widehat{\mathbf{s}}^{(i)} = \rho'(\widehat{\mathbf{h}}_G^{(i)}) \in \mathbb{R}^K. \quad (3)$$

Here, $\sigma(\cdot)$ denotes the softmax function, $\rho'(\cdot) : \mathbb{R}^F \to \mathbb{R}^C$ represents the clustering classification head, which takes $\widehat{\mathbf{h}}_G$ from the GNN encoder $h(\cdot)$ as input. The reweighting coefficient for each sample $G^{(i)}$ is denoted as $w^{(i)}$, which adjusts the weight for samples from the majority group by reducing it and increases the weight for samples from the minority group. The generalized cross-entropy (GCE) method (Zhang & Sabuncu, 2018) is employed to calculate $w^{(i)}$, i.e., $w^{(i)} = \frac{1-\left(\mathbf{s}_j^{(i)}\right)^\gamma}{\gamma}$, where $j$ is the ground-truth clustering label for sample $G^{(i)}$, and $\gamma$ is a hyperparameter. Finally, The loss objective for feature disentanglement is:

$$\mathcal{L} = \mathcal{L}_{GT} + \lambda \mathcal{L}_{inv}, \quad (4)$$

where $\mathcal{L}_{GT}$ denotes the ERM loss. $\mathcal{L}_{inv}$, serving as a regularization term, will guide the learning process to learn invariant features by removing spurious features from the ERM-learned features. Formally, we present the following theorem.

**Theorem 4.2.** *There exists a suitable $\gamma$ and clustering number $K$, such that minimizing the loss objective $\mathcal{L} = \mathcal{L}_{GT} + \lambda \mathcal{L}_{inv}$ will lead to the optimal encoder $h^*(\cdot)$ which elicits invariant features for any graph $G$, i.e., $\widehat{\mathbf{h}}_G = h^*(G) = \mathbf{h}_c$.*

The proof is included in Appendix D. In the proof of Theorem 4.2, we show that $\mathcal{L}_{inv}$ will guide the ERM-learned features to focus on invariant features by removing or unlearning the spurious ones. More concretely, given ERM learns both invariant and spurious features (Kirichenko et al., 2023; Chen et al., 2023b), i.e., $\widehat{\mathbf{h}}_G = \kappa(\mathbf{h}_c, \mathbf{h}_s)$, where $\kappa(\cdot)$ is a functional mapping to graph representation $\widehat{\mathbf{h}}_G$ given a set of invariant features $\mathbf{h}_c$ and a set of spurious features $\mathbf{h}_s$, $\mathcal{L}_{inv}$ regularizes the learning

Table 1: Performance on synthetic and real-world datasets. Numbers in **bold** indicate the best performance, while the underlined numbers indicate the second best performance.

| Method | GOOD-Motif | | GOOD-HIV | | OGBG-Molbace | | OGBG-Molbbbp | |
|---|---|---|---|---|---|---|---|---|
| | base | size | scaffold | size | scaffold | size | scaffold | size |
| ERM | 68.66$_{\pm4.25}$ | 51.74$_{\pm2.88}$ | 69.58$_{\pm2.51}$ | 59.94$_{\pm2.37}$ | 75.11$_{\pm3.03}$ | 83.60$_{\pm3.47}$ | 68.10$_{\pm1.68}$ | 78.29$_{\pm3.76}$ |
| IRM | 70.65$_{\pm4.17}$ | 51.41$_{\pm3.78}$ | 67.97$_{\pm1.84}$ | 59.00$_{\pm2.92}$ | 75.47$_{\pm2.22}$ | 83.12$_{\pm2.58}$ | 67.22$_{\pm1.15}$ | 77.56$_{\pm2.48}$ |
| GroupDRO | 68.24$_{\pm8.92}$ | 51.95$_{\pm5.86}$ | 70.64$_{\pm2.57}$ | 58.98$_{\pm2.16}$ | - | - | 66.47$_{\pm2.39}$ | 79.27$_{\pm2.43}$ |
| VREx | 71.47$_{\pm6.69}$ | 52.67$_{\pm5.54}$ | 70.77$_{\pm2.84}$ | 58.53$_{\pm2.88}$ | 72.81$_{\pm4.29}$ | 82.55$_{\pm2.51}$ | 68.74$_{\pm1.03}$ | 78.76$_{\pm2.37}$ |
| RSC | 46.12$_{\pm3.76}$ | 51.70$_{\pm5.47}$ | 69.16$_{\pm3.23}$ | 61.17$_{\pm0.74}$ | 74.59$_{\pm3.65}$ | 84.34$_{\pm2.65}$ | 69.01$_{\pm2.84}$ | 78.07$_{\pm3.89}$ |
| DiverseModel | 54.24$_{\pm8.22}$ | 41.01$_{\pm1.98}$ | 69.17$_{\pm3.62}$ | 61.59$_{\pm2.23}$ | 73.48$_{\pm3.56}$ | 79.40$_{\pm1.70}$ | 68.04$_{\pm3.27}$ | 77.62$_{\pm1.90}$ |
| DropEdge | 45.08$_{\pm4.46}$ | 45.63$_{\pm4.61}$ | 70.78$_{\pm1.38}$ | 58.53$_{\pm1.26}$ | 70.81$_{\pm2.12}$ | 76.39$_{\pm2.29}$ | 66.49$_{\pm1.55}$ | 78.32$_{\pm3.44}$ |
| FLAG | 61.12$_{\pm5.39}$ | 51.66$_{\pm4.14}$ | 68.45$_{\pm2.30}$ | 60.59$_{\pm2.95}$ | 80.37$_{\pm1.58}$ | 84.72$_{\pm0.88}$ | 67.69$_{\pm2.36}$ | 79.26$_{\pm2.26}$ |
| LiSA | 54.59$_{\pm4.81}$ | 53.46$_{\pm3.41}$ | 70.38$_{\pm1.45}$ | 52.36$_{\pm3.73}$ | 78.05$_{\pm5.01}$ | 83.92$_{\pm2.52}$ | 68.11$_{\pm0.52}$ | 78.62$_{\pm3.74}$ |
| DIR | 62.07$_{\pm8.75}$ | 52.27$_{\pm4.56}$ | 68.07$_{\pm2.29}$ | 58.08$_{\pm2.31}$ | 75.49$_{\pm2.80}$ | 77.42$_{\pm7.43}$ | 66.86$_{\pm2.25}$ | 76.40$_{\pm4.43}$ |
| DisC | 51.08$_{\pm3.08}$ | 50.39$_{\pm1.15}$ | 68.07$_{\pm1.75}$ | 58.76$_{\pm0.91}$ | 57.78$_{\pm3.60}$ | 71.13$_{\pm8.86}$ | 67.12$_{\pm2.11}$ | 56.59$_{\pm10.09}$ |
| CAL | 65.63$_{\pm4.29}$ | 51.18$_{\pm5.60}$ | 67.37$_{\pm3.61}$ | 57.95$_{\pm2.24}$ | 76.29$_{\pm1.60}$ | 79.68$_{\pm4.06}$ | 68.06$_{\pm2.60}$ | 79.50$_{\pm4.81}$ |
| GREA | 56.74$_{\pm9.23}$ | 54.13$_{\pm10.02}$ | 67.79$_{\pm2.56}$ | 60.71$_{\pm2.20}$ | 77.16$_{\pm1.37}$ | 83.15$_{\pm9.07}$ | 69.72$_{\pm1.66}$ | 77.34$_{\pm3.52}$ |
| GSAT | 62.80$_{\pm11.41}$ | 53.20$_{\pm8.35}$ | 68.66$_{\pm1.35}$ | 58.06$_{\pm1.98}$ | 72.32$_{\pm5.66}$ | 82.45$_{\pm2.73}$ | 66.78$_{\pm1.45}$ | 75.63$_{\pm3.83}$ |
| CIGA | 66.43$_{\pm11.31}$ | 49.14$_{\pm8.34}$ | 69.40$_{\pm2.39}$ | 59.55$_{\pm2.56}$ | 76.44$_{\pm1.72}$ | 83.95$_{\pm2.75}$ | 64.92$_{\pm2.09}$ | 65.98$_{\pm3.31}$ |
| AIA | 73.64$_{\pm5.15}$ | 55.85$_{\pm7.98}$ | 71.15$_{\pm1.81}$ | 61.64$_{\pm3.37}$ | 79.42$_{\pm2.01}$ | 85.11$_{\pm0.74}$ | 70.79$_{\pm1.53}$ | 81.03$_{\pm5.15}$ |
| OOD-GCL | 56.46$_{\pm4.61}$ | 60.23$_{\pm8.49}$ | 70.85$_{\pm2.07}$ | 58.48$_{\pm2.94}$ | 75.96$_{\pm2.21}$ | 85.34$_{\pm1.77}$ | 67.28$_{\pm3.09}$ | 78.11$_{\pm3.32}$ |
| EQuAD | 67.11$_{\pm10.11}$ | 59.72$_{\pm3.69}$ | 72.24$_{\pm0.64}$ | 64.19$_{\pm0.56}$ | 79.15$_{\pm2.32}$ | 86.41$_{\pm5.63}$ | 70.22$_{\pm2.36}$ | 80.82$_{\pm5.28}$ |
| LIRS | **75.51**$_{\pm2.19}$ | **74.95**$_{\pm7.69}$ | **72.82**$_{\pm1.61}$ | **66.64**$_{\pm1.44}$ | **81.91**$_{\pm1.98}$ | **88.77**$_{\pm1.64}$ | **71.04**$_{\pm0.76}$ | **82.19**$_{\pm1.57}$ |

process to optimize $\kappa^*(\cdot)$ such that $\widehat{\mathbf{h}}_G$ contains only $\mathbf{h}_c$. While previous OOD methods achieve a similar goal (to learn only invariant features), our approach differs by first learning the spurious features and then leveraging them to identify the causal features from the ERM-learned features, a process we refer to as *spurious feature removing or unlearning.*

## 5 EXPERIMENTS

In this section, we perform empirical study on both synthetic and real-world datasets. More details about the datasets and experiment setup are included in Appendix H.

### 5.1 EXPERIMENTAL SETUP

**Datasets.** We adopt GOODMotif and GOODHIV datasets (Gui et al., 2022), OGBG-Molbace and OGBG-Molbbbp datasets (Hu et al., 2020; Wu et al., 2018) to comprehensively evaluate the OOD generalization performance of our proposed framework. For GOODMotif datasets, we adopt base shift and size shift, for OGBG datasets, in addition to scaffold shift, we also create size shift, following previous studies (Gui et al., 2022; Sui et al., 2023). More details on the datasets used in our work are included in Appendix H.

**Baselines.** Besides ERM (Vapnik, 1995), we compare our method against three lines of OOD baselines: (1) OOD algorithms on Euclidean data, including IRM (Arjovsky et al., 2020), VREx (Krueger et al., 2021), and GroupDRO (Sagawa et al., 2019); (2) Diverse feature learning methods, including RSC (Huang et al., 2020), and DiverseModel (Teney et al., 2022). (3) graph-specific OOD and data augmentation algorithms without requiring environment labels, including DIR (Wu et al., 2022c), GSAT (Miao et al., 2022), GREA (Liu et al., 2022), DisC (Fan et al., 2022), CIGA (Chen et al., 2022), AIA (Sui et al., 2023), OOD-GCL (Li et al., 2024a), EQuAD (Yao et al., 2024), DropEdge (Rong et al., 2019), FLAG (Kong et al., 2022), and LiSA (Yu et al., 2023). Details of the baselines and their implementation are provided in Appendix H.

**Evaluation.** We report the ROC-AUC score for GOOD-HIV, OGBG-Molbbbp, and OGBG-Molbace datasets, where the tasks are binary classification. For GOOD-Motif and SPMotif datasets, we use accuracy as the evaluation metric. We run experiments 4 times with different random seeds, select models based on the validation performance, and report the mean and standard deviations on the test set.

## 5.2 EXPERIMENTAL RESULTS

In this section, we report the main results on both synthetic and real-world datasets.

**Synthetic datasets.** LIRS achieves superior performance on GOOD-Motif, surpassing the second-best method by $12.51\%$ (base split) and $25.50\%$ (size split). This highlights its effectiveness in capturing domain-invariant features. While EQuAD also removes spurious features, its infomax-based learning may inadvertently retain them, whereas LIRS's biased infomax effectively suppresses spurious signals. OOD-GCL (Li et al., 2024a) employs contrastive learning without labeled data, making invariant feature learning more challenging. As shown in Table 1, LIRS outperforms OOD-GCL across all datasets, benefiting from labeled data during training. We also compare LIRS with RSC Huang et al. (2020) and DiverseModel Teney et al. (2022), which aim to learn diverse features via ERM. As illustrated in Table 1, although RSC and DiverseModel perform well on real-world datasets, they achieve suboptimal results on the GOOD-Motif datasets. This discrepancy may arise from the fact that these methods attempt to capture diverse features, while in the GOOD-Motif datasets, only one invariant subgraph is causally related to the target. As a result, these methods may mistakenly capture non-generalizable patterns in $G_s$. In contrast, LIRS leverages biased infomax to effectively learn spurious patterns and subsequently identify the invariant subgraph by unlearning the spurious patterns. Lastly, LIRS significantly outperforms other OOD methods that aim to directly learn invariant features, showcasing the efficacy of the proposed learning paradigm.

**Real-world datasets.** In real-world datasets, which present more complex and realistic distribution shifts, many graph OOD algorithms exhibit instability, occasionally underperforming ERM. In contrast, LIRS adopts an indirect learning paradigm, which may enable the learning of more invariant features compared to previous graph invariant learning methods, leading to more effective OOD generalization. Similarly to the results on synthetic datasets, LIRS also demonstrates superior OOD performance under both scaffold shift and size shift compared to other methods. EQuAD, which adopts a similar learning paradigm as LIRS, shows a comparable trend, with a more significant advantage under the size shift relative to other methods. Furthermore, the improvements of LIRS over EQuAD across various datasets highlights the effectiveness of the biased infomax principle and the intra-class cross-entropy loss for learning spurious features and invariant features respectively. Compared to other baselines that learns invariant features directly, LIRS outperforms the best method (AIA) by $4.62\%$ in size shift and $1.95\%$ in scaffold shift respectively.

## 5.3 IN-DEPTH ANALYSIS

**Can LIRS learn more invariant features?** To further validate the quality of invariant features learned by LIRS, we curated a dataset derived from the SPMotif dataset. Specifically, we constructed a binary classification dataset where the motifs *House* and *Crane* corresponds to label 0, and *Diamond* and *Cycle* corresponds to label 1. During the construction of each class's samples, we attached both invariant subgraphs to the base subgraph with $50\%$ chance, while in the remaining $50\%$, we randomly attached one invariant subgraph to the base spurious subgraph. For the test set, we randomly attached a single invariant subgraph to the base subgraph. Similar to the SPMotif dataset, the base spurious subgraph was correlated with the target labels, maintaining an equal correlation strength with the labels in the test set. In addition to evaluating LIRS against traditional methods such as IRM, VRex and their variants, we also incorporated strong graph-specific OOD baselines, including CIGA, AIA, and EQuAD for comparisons. The experimental results in Table 2 demonstrate that LIRS consistently outperforms other baselines across varying spurious correlation strengths, indicating its superior ability to learn more invariant features. Similarly, EQuAD, adopting a similar learning paradigm, also achieves competitive performance in SPMotif-binary datasets. This demonstrates that learning invariant features indirectly can lead to

Table 2: Performance on the SPMotif-binary datasets under varying spurious correlations.

| Method | SPMotif-binary | | |
|---|---|---|---|
| | $b = 0.40$ | $b = 0.60$ | $b = 0.90$ |
| ERM | $74.93_{\pm 3.94}$ | $72.78_{\pm 3.15}$ | $63.78_{\pm 4.18}$ |
| IRM | $76.01_{\pm 4.12}$ | $70.85_{\pm 4.73}$ | $66.55_{\pm 4.80}$ |
| VREx | $79.03_{\pm 1.02}$ | $73.78_{\pm 1.75}$ | $65.27_{\pm 6.78}$ |
| GIL | $77.15_{\pm 3.18}$ | $73.85_{\pm 2.76}$ | $68.90_{\pm 7.28}$ |
| GREA | $79.65_{\pm 6.36}$ | $73.01_{\pm 7.99}$ | $69.85_{\pm 0.35}$ |
| CIGA | $76.93_{\pm 3.94}$ | $71.70_{\pm 1.55}$ | $66.80_{\pm 5.35}$ |
| AIA | $78.46_{\pm 3.19}$ | $71.83_{\pm 0.69}$ | $64.37_{\pm 4.14}$ |
| EQuAD | $80.82_{\pm 0.65}$ | $74.20_{\pm 4.10}$ | $69.79_{\pm 7.81}$ |
| LIRS | $\mathbf{82.17_{\pm 0.91}}$ | $\mathbf{75.32_{\pm 1.65}}$ | $\mathbf{71.29_{\pm 2.12}}$ |

learning a more comprehensive set of invariant features and achieving better OOD generalization ability.

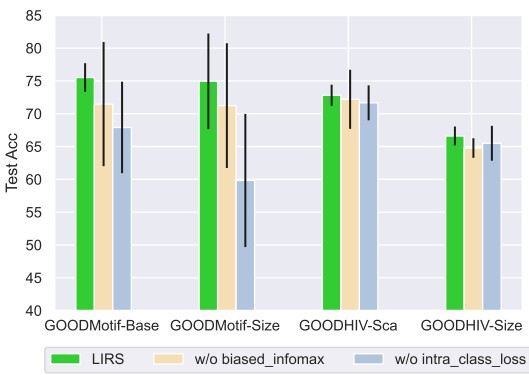

(a) Ablation study on biased infomax and intra-class cross-entropy loss.

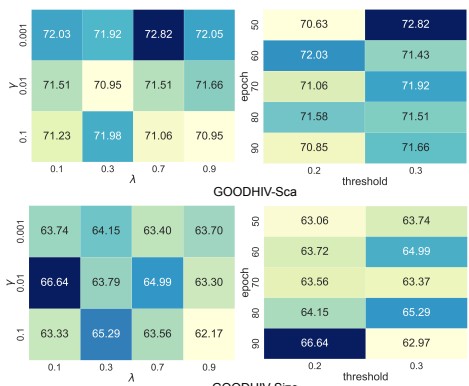

(b) Hyperparameter sensitivity analysis on GOODHIV datasets.

Figure 4: Ablation and hyperparameter sensitivity studies for LIRS.

**Ablation study.** We conducted an ablation study on biased infomax and class-conditioned (intra-class) cross-entropy loss to evaluate their effectiveness in LIRS. As illustrated in Figure 4a, replacing biased infomax with vanilla infomax to generate spurious features leads to performance degradation across all datasets. This decline is primarily due to the enhanced capability of biased infomax in learning spurious features, which is crucial for subsequent steps for learning graph invariance. Furthermore, replacing class-conditioned cross-entropy loss with standard cross-entropy loss also led to negative effects, which is mainly due to the interference of spurious patterns across different classes, as discussed in Sec. 4. One typical empirical support arises from the Motif-size dataset, where each spurious pattern is correlated with different target labels with nearly equal strength. As a result, even though biased infomax can generate accurate spurious features, the logits generated using the standard cross-entropy loss are not sufficiently precise, which aligns with Prop. 2. This limitation significantly constrains the test performance, as illustrated in Figure 4a. In contrast, the class-conditioned cross-entropy loss is able to generate more accurate spurious logits, thus enhancing the OOD performance.

**Hyperparameter Sensitivity.** We investigate the impact of hyperparameters in LIRS, including the reweighting coefficient $\gamma$, the penalty weights $\lambda$, and hyperparameters related to biased infomax. Specifically, we analyze the effect of the epoch $E$ at which embeddings are derived from biased infomax and the threshold used to determine whether a sample should be considered biased or not. As illustrated in Figure 4b, for both the GOODHIV-Sca and GOODHIV-Size datasets, LIRS demonstrates stable performance, highlighting its robustness to variations in hyperparameter settings. This consistency suggests that LIRS is agnostic to hyperparameter tuning, which further emphasizes its applicability in real-world scenarios.

**How do different size split methods affect the OOD performance?** For the real-world datasets, the size split was performed in descending order, where the graphs in the training set have a larger number of nodes compared to those in the test set. To evaluate the impact of varying graph sizes on generalization ability of various OOD methods, we also performed size splits based on ascending order for the OGBG-Molbace and OGBG-Molbbbp datasets. The results of these experiments are presented in Table 3. As shown, LIRS continues to outperform all competitive baselines under this alternative size split. Specifically, LIRS achieves the highest ROC-AUC scores on both OGBG-Molbace and OGBG-

Table 3: Performance on OGBG datasets using a different size split procedure, where the graphs in the training set have a smaller number of nodes compared to those in the test set.

| Method | OGBG Datasets | |
| --- | --- | --- |
| | OGBG-Molbace | OGBG-Molbbbp |
| ERM | $76.31_{\pm 0.58}$ | $87.31_{\pm 2.37}$ |
| IRM | $79.44_{\pm 3.05}$ | $88.77_{\pm 2.45}$ |
| VREx | $76.38_{\pm 1.75}$ | $84.20_{\pm 2.80}$ |
| GREA | $77.46_{\pm 5.57}$ | $86.18_{\pm 2.54}$ |
| GSAT | $72.29_{\pm 4.45}$ | $87.46_{\pm 2.67}$ |
| CIGA | $77.89_{\pm 3.68}$ | $87.94_{\pm 0.86}$ |
| AIA | $78.62_{\pm 2.88}$ | $89.18_{\pm 1.77}$ |
| EQuAD | $82.34_{\pm 3.42}$ | $88.65_{\pm 3.83}$ |
| LIRS | $\mathbf{84.62}_{\pm \mathbf{0.84}}$ | $\mathbf{90.24}_{\pm \mathbf{4.14}}$ |

Molbbbp datasets. This highlights the superiority of LIRS in generalizing to larger graphs when trained on smaller ones, further confirming its potential for real-world applications.

## 6    RELATED WORK

**Learning invariant features.** OOD generalization is a critical challenge in machine learning, where models trained on a specific data distribution often fail to generalize well to unseen distributions. Recently invariance learning has been proposed to tackle this issue, which builds upon the theory of causality (Peters et al., 2016; Pearl, 2009) to learn causally-related representation that remain stable across different environments (Arjovsky et al., 2020; Parascandolo et al., 2020; Mahajan et al., 2021; Wald et al., 2021; Ahuja et al., 2020; 2021). Inspired from IRM (Arjovsky et al., 2020), several invariant learning methods on graphs are proposed (Yang et al., 2022; Li et al., 2022b; Fan et al., 2022; Liu et al., 2022; Wu et al., 2022c; Chen et al., 2022; 2023a; Gui et al., 2023; Sui et al., 2023; Li et al., 2024b) to learn graph-level representations that are robust to distribution shifts. Most of these methods aim to learn invariant features directly by training an equipredictive classifier. Recent study (Yao et al., 2024) has proposed a new learning paradigm to learn graph invariance indirectly by learning spurious features first, then disentangle them from the ERM-learned representations. In this work, we adopt this learning paradigm for learning graph invariance.

**Self-supervised learning induces spuriosity.** Self-supervised learning (SSL) has emerged as a potent paradigm for learning representations from unlabeled datasets. The fundamental concept involves devising pretext tasks that maximize similarity between two augmented views of the same data point, often utilizing the InfoNCE loss (Oord et al., 2018). This approach has been widely applied in domains such as images (Chen et al., 2020a;b) and graphs (Veličković et al., 2019; Sun et al., 2019; Zhu et al., 2020; You et al., 2020). Recent work have shown that SSL may tend to learn spurious features in both image domain (Hamidieh et al., 2024; Meehan et al., 2023) and graph domain (Yao et al., 2024). Hamidieh et al. (2024) attempted to enhance SSL model performance by addressing this issue and promoting the learning of invariant features. In contrast, our goal in this work is to improve SSL model ability to capture spuriosity, thereby facilitating better feature disentanglement for invariant learning.

**Learning diverse features.** Recent studies have shown that ERM tends to encourage models to learn the simplest predictive features (Hermann & Lampinen, 2020; Kalimeris et al., 2019; Neyshabur et al., 2014; Pezeshki et al., 2021). This simplicity bias causes the models to rely on simple (spurious) but non-causal features, ignoring more complex patterns that might be equally predictive. To address this challenge, Huang et al. (2020) employs a self-challenging mechanism to force the model to learn diverse patterns by discarding dominant features, while Teney et al. (2022) constructs diverse features by training a collection of classifiers with diversity regularization. Additionally, Zhang et al. (2022a); Chen et al. (2023b) adopt DRO to iteratively explore new features. While previous works aim to encourage models to learn a diverse set of predictive patterns, our method takes a different approach. Specifically, we aim to remove or unlearn spurious features from ERM-learned features, thereby allowing the model to capture more invariant features.

## 7    CONCLUSIONS

In this study, we identified a critical limitation in IRM, VRex and their variants on graph invariant learning, i.e., these methods may only capture a subset of the invariant features, thereby limiting their OOD generalization performance. To address this issue, we investigate the effectiveness of learning invariant features *indirectly* by first learning and then removing spurious features. Our theoretical and empirical analyses demonstrate that this approach facilitates the learning of a more comprehensive set of invariant features than traditional (graph) invariant learning methods, thereby improving generalization to unseen environments. We then propose **LIRS** that adopts this learning paradigm, which consists of: a) The biased infomax principle, and b) The class-conditioned cross-entropy loss, which elicit effective spuriosity learning and invariant learning respectively, aiming to learn more invariant features. Extensive experiments on both synthetic and real-world datasets demonstrate the superiority of our proposed method over existing state-of-the-art OOD algorithms.

ACKNOWLEDGEMENT

We would like to acknowledge the support from NSF Award No.2229881, the AI Institute for Societal Decision Making (AI-SDM), the National Institutes of Health (NIH) under Contract R01HL159805, as well as grants from Quris AI, Florin Court Capital, and the MBZUAI-WIS Joint Program.

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

# APPENDIX

## A  NOTATIONS

We present a set of notations used throughout our paper for clarity. Below are the main notations along with their definitions.

Table 4: Notation Table

| Symbols | Definitions |
|---|---|
| $\mathcal{G}$ | a set of graphs |
| $G = (A, X)$ | a graph with the adjacency matrix $A \in \{0,1\}^{n \times n}$ and node feature matrix $X \in \mathbb{R}^{n \times d}$ |
| $Y$ | random variable for labels |
| $C$ | content factor |
| $S$ | style factor |
| $E$ | environment |
| $\mathbf{h}_c$ | invariant representations (features) |
| $\mathbf{h}_s$ | spurious representations (features) |
| $G_c$ | the invariant subgraph with respect to $G$ |
| $G_s$ | the spurious subgraph with respect to $G$ |
| $\widehat{G}_c$ | the estimated invariant subgraph |
| $\widehat{G}_s$ | the estimated spurious subgraph |
| $\widehat{\mathbf{h}}_G$ | the estimated graph representation for graph $G$ |
| $\widehat{\mathbf{h}}_i$ | the estimated node representation for node $i \in G$ |
| $\widehat{\mathbf{h}}_c$ | the estimated invariant graph representation, interchangeably with $\widehat{C}$ in our paper |
| $\widehat{\mathbf{h}}_s$ | the estimated spurious graph representation, interchangeably with $\widehat{S}$ in our paper |
| $\mathbf{s}_j^{(i)}$ | The $j^{th}$ entry of the spurious logits derived from $\mathbf{h}_s^{(i)}$ for sample $G^{(i)}$ |
| $\widehat{\mathbf{s}}_j^{(i)}$ | The $j^{th}$ of the estimated spurious logits for sample $G^{(i)}$ |
| $h(\cdot)$ | encoder |
| $\rho(\cdot)$ | classification head for $\mathcal{L}_{GT}$ |
| $\rho'(\cdot)$ | classification head for $\mathcal{L}_{Inv}$ |
| $w^{(i)}$ | reweighting coefficient for sample $G^{(i)}$ |
| $[K] := \{1, 2, \cdots, K\}$ | index set with $K$ elements |
| $\mathbf{1}_K$ | all-one (column) vector with $K$ entries |
| $\mathbf{w}$ | a vector |
| $\mathbf{W}$ | a matrix |
| $W$ | a random variable |
| $\mathcal{W}$ | a set |

## B  MORE BACKGROUND AND PRELIMINARIES

**Graph Neural Networks.** In this work, we adopt message-passing GNNs for graph classification due to their expressiveness. Given a simple and undirected graph $G = (\mathbf{A}, \mathbf{X})$ with $n$ nodes and $m$ edges, where $\mathbf{A} \in \{0,1\}^{n \times n}$ is the adjacency matrix, and $\mathbf{X} \in \mathbb{R}^{n \times d}$ is the node feature matrix with $d$ feature dimensions, the graph encoder $h : \mathbb{G} \to \mathbb{R}^h$ aims to learn a meaningful graph-level representation $h_G$, and the classifier $\rho : \mathbb{R}^h \to \mathbb{Y}$ is used to predict the graph label $\widehat{Y}_G = \rho(h_G)$. To obtain the graph representation $h_G$, the representation $\mathbf{h}_v^{(l)}$ of each node $v$ in a graph $G$ is iteratively updated by aggregating information from its neighbors $\mathcal{N}(v)$. For the $l$-th layer, the updated representation is obtained via an AGGREGATE operation followed by an UPDATE operation:

$$\mathbf{m}_v^{(l)} = \text{AGGREGATE}^{(l)}\left(\left\{\mathbf{h}_u^{(l-1)} : u \in \mathcal{N}(v)\right\}\right), \tag{5}$$

$$\mathbf{h}_v^{(l)} = \text{UPDATE}^{(l)}\left(\mathbf{h}_v^{(l-1)}, \mathbf{m}_v^{(l)}\right), \tag{6}$$

where $\mathbf{h}_v^{(0)} = \mathbf{x}_v$ is the initial node feature of node $v$ in graph $G$. Then GNNs employ a READOUT function to aggregate the final layer node features $\left\{\mathbf{h}_v^{(L)} : v \in \mathcal{V}\right\}$ into a graph-level representation $\mathbf{h}_G$:

$$\mathbf{h}_G = \text{READOUT}\left(\left\{\mathbf{h}_v^{(L)} : v \in \mathcal{V}\right\}\right). \tag{7}$$

**Data Generating Process.** The presumed data generating process adopted in this work is similar to the previous studies (Arjovsky et al., 2020; Ahuja et al., 2020; Chen et al., 2022; 2023a; Wu et al., 2022a). The dynamics of graph generation are fundamentally grounded upon the principles of Structural Causal Models (SCM) (Peters et al., 2016), capturing the complex interplay between latent variables and their influence on observable graph characteristics. This model assumes the graph generation as a function $f_{\text{gen}} : \mathbb{Z} \to \mathbb{G}$, where $\mathbb{Z} \subseteq \mathbb{R}^n$ denotes the latent space, and $\mathbb{G}$ is the graph space. Through the lens of SCM, the generation process is decomposed into the invariant subgraph $G_c$, the spurious subgraph $G_s$, and the observed graph $G$ respectively, as formulated in the following equation:

$$G_c := f_{\text{gen}}^{G_c}(C), \quad G_s := f_{\text{gen}}^{G_s}(S), \quad G := f_{\text{gen}}^{G}(G_c, G_s).$$

Furthermore, the latent interactions among $C$, $S$, and $Y$ can exhibit two types of conditional independence: (i) $Y \perp\!\!\!\perp S \mid C$ and (ii) $Y \not\!\perp\!\!\!\perp S \mid C$. In case (i), the invariant factor $C$ is fully informative (FIIF) to the target label $Y$, and the latent spurious factor $S$ provide no further information. In case (ii), the invariant factor $C$ is only partially informative (PIIF) about $Y$, spurious factor $S$ can further provide additional information to aid the prediction of $Y$, however, as $S$ is directly affected by $E$, it is not stable across different environments. The SCMs for the two scenarios are illustrated in Figure 5. For the PIIF SCM, the structure equation is:

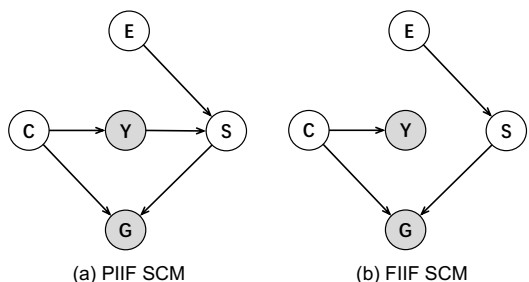

Figure 5: Structure causal models for graph data generation.

$$Y = X_c + n_1, X_s = Y + n_2 + \epsilon^e, \tag{8}$$

while for FIIF SCM, the structure equation is:

$$Y = X_c + n_1, X_s = n_2 + \epsilon^e. \tag{9}$$

Here $X_c$ and $X_s$ denote the random variables for invariant factor and spurious factor respectively. $n_1$ and $n_2$ represent random Gaussian noise, and $\epsilon^e$ stands for an environmental variable, which causes the spurious correlation between $X_s$ and $Y$.

## C ADDITIONAL RELATED WORK

**OOD Generalization.** OOD generalization is a critical challenge in machine learning, where models trained on a specific data distribution often fail to generalize well to unseen distributions. Several approaches have been proposed to address this issue, including domain generalization, distributional robustness optimization (DRO), and invariance learning. *Domain generalization* aims to learn features that are invariant across different domains or environments. Previous studies, such as Ganin et al. (2016); Sun & Saenko (2016); Li et al. (2018); Dou et al. (2019), regularize the learned features to be domain-invariant. *DRO* methods focus on training models to perform robust against the worst-case scenarios among diverse data groups. Namkoong & Duchi (2016); Hu et al. (2018); Sagawa et al. (2019) regularize models to be robust to mild distributional perturbations of the training distributions, expecting the models to perform well in unseen test environments. Building upon this, Liu et al. (2022) Zhang et al. (2022b) and Yao et al. (2022) propose advanced strategies to improve robustness by assuming that models trained with ERM have a strong reliance on spurious features.

*Invariance learning* leverages the theory of causality (Peters et al., 2016; Pearl, 2009) and introduces causal invariance to the learned representations. The Independent Causal Mechanism (ICM) assumption in causality states that the conditional distribution of each variable given its causes does not inform or influence other conditional distributions. Despite changes to the intervened variables, the conditional distribution of intervened variables and the target variable remains invariant. Arjovsky et al. (2020) proposes the framework of Invariant Risk Minimization (IRM) that allows the adoption of causal invariance in deep neural networks, inspiring various invariant learning works such as Parascandolo et al. (2020); Mahajan et al. (2021); Wald et al. (2021); Ahuja et al. (2020; 2021); Krueger et al. (2021).

**Graph-Level OOD Generalization.** Recently, there has been a growing interest in learning graph-level representations that are robust under distribution shifts, particularly from the perspective of invariant learning. MoleOOD Yang et al. (2022) and GIL Li et al. (2022b) propose to infer environmental labels to assist in identifying invariant substructures within graphs. DIR Wu et al. (2022c), GREA Liu et al. (2022) and iMoLD Zhuang et al. (2023) employ environment augmentation techniques to facilitate the learning of invariant graph-level representations. These methods typically rely on the explicit manipulation of unobserved environmental variables to achieve generalization across unseen distributions. AIA Sui et al. (2023) employs an adversarial augmenter to explore OOD data by generating new environments while maintaining stable feature consistency. To circumvent the need for environmental inference or augmentation, CIGA Chen et al. (2022) and GALA Chen et al. (2023a) utilizes supervised contrastive learning to identify invariant subgraphs based on the assumption that samples sharing the same label exhibit similar invariant subgraphs. EQuAD Yao et al. (2024) adopts self-supervised learning to learn spriosu efatures first, followed by learning invariant features by unlearning spurious features. LECI Gui et al. (2023) and G-Splice Li et al. (2023b) assume the availability of environment labels, and study environment exploitation strategies for graph OOD generalization. LECI Gui et al. (2023) proposes to learn a causal subgraph selector by jointly optimizing label and environment causal independence, and G-Splice Li et al. (2023b) studies graph and feature space extrapolation for environment augmentation, which maintains causal validity. On the other hand, some works do not utilize the invariance principle for graph OOD generalization. DisC Fan et al. (2022) initially learns a biased graph representation and subsequently focuses on unbiased graphs to discover invariant subgraphs. GSAT Miao et al. (2022) utilizes information bottleneck principle Tishby & Zaslavsky (2015) to learn a minimal sufficient subgraph for GNN explainability, which is shown to be generalizable under distribution shifts. OOD-GNN Li et al. (2022a) proposes to learn disentangled graph representation by computing global weights of all data. Parallel to all previous studies, we propose ELiSD, which utilizes spurious subgraph diversification to provably identify $G_c$ for OOD generalization, and uplift the lower bound of $I(G; Y)$ to enhance feature learning simultaneously. In this study, we adopt the same learning paradigm as EQuAD, and we propose instance-level adaptive biased infomax and intra-class cross entropy loss to enhance the efficacy of learning spurious and invariant features respectively.

**Node-Level OOD Generalization.** There has also been a substantial amount of work focusing on the OOD generalization problem on node-level classification tasks. Most of these studies (Wu et al., 2022a; Liu et al., 2023; Li et al., 2023a; Yu et al., 2023) also focus on environment generation, which facilitates the subsequent invariant learning. These methods are also likely to learn only a subset of invariant features, thus limiting their ability to generalize to OOD data.

**GNN Explainability.** GNN explanation methods can be broadly categorized into instance-level (Luo et al., 2020; Ying et al., 2019; Pope et al., 2019; Schnake et al., 2021) and model-level (Azzolin et al., 2022; Yuan et al., 2020; Wang & Shen, 2022) explanations. The main goal of the instance-level explanation methods is to explain why a certain prediction is made for a particular instance. Specifically, the important input subgraph and features that contribute the most to model predictions are identified for a given instance. In this work, we employ instance-level GNN explanation methods to approximate the invariant subgraph $G_c$, and utilize off-the-shelf method GSAT (Miao et al., 2022), that is robust under distribution shifts, to identify the important nodes in a graph to facilitate the learning of graph spuriosity through adaptive biased infomax in real-world datasets.

# D THEORETICAL RESULTS

## D.1 THEORETICAL DISCUSSION ON VREX IN LEARNING INVARIANT FEATURES

We first outline the objective function of VRex as following:

$$\mathcal{L}_{VRex} := \min_{w,\phi} \mathbb{E}_e\left[\mathcal{L}\left(w \circ \phi\left(X\right), Y\right)\right] + \beta \operatorname{Var}_e\left[\mathcal{L}\left(w \circ \phi\left(X\right), Y\right)\right], \tag{10}$$

here $w$ and $\phi$ denote the classifier and feature extractor, respectively. $\beta$ is the hyperparameter to control regularization strength. Next we prove the following proposition.

**Proposition 3.** *Let* $w^*, \phi^* = \underset{w,\phi}{\operatorname{argmin}}\mathcal{L}_{VRex}$, *then* $\widehat{\mathbf{h}}_G = w^*(X)$ *learns invariant features, however* $\widehat{\mathbf{h}}_G$ *may only contain a subset of invariant features, given our PIIF data generating assumption:*

$$Y = X_c + n_1, X_s = Y + n_2 + \epsilon^e, \text{ where } X_c := \sum_i X_{c,i}, X_s := \sum_i X_{s,i}, \tag{11}$$

*and FIIF data generating assumption:*

$$Y = X_c + n_1, X_s = n_2 + \epsilon^e, \text{ where } X_c := \sum_i X_{c,i}, X_s := \sum_i X_{s,i}, \tag{12}$$

*when using a linear classifier* $\phi(X) = w_1 X_c + w_2 X_s$, *and* $\mathcal{L}(w \circ \phi(X)) := R(e) = (\phi(X) - Y)^2$.

Here, we assume an additive model for $X_c$, i.e., A set of invariant features $X_{c,i}$ is combined through summation to obtain the final $X_c$, and similarly for $X_s$.

*Proof.* To prove Prop. 3, we first expand $\operatorname{Var}_e(R(e)) = \mathbb{E}_e\left[R^2(e)\right] - (\mathbb{E}_e[R(e)])^2$, then we take derivative w.r.t. $w_1$ and $w_2$, we then show that Eqn. 10 can elicit $w^*$ that learns invariant features, however, it may only learn a subset of them.

Given $R(e) = (\phi(X) - Y)^2$ and $\operatorname{Var}_e(R(e)) = \mathbb{E}_e R^2(e) - \mathbb{E}_e^2 R(e)$, we have:

$$\begin{aligned}
\frac{\partial \operatorname{Var}_e(R(e))}{\partial w_1} &= \frac{\partial \mathbb{E}_e[(\phi(X) - Y)^4]}{\partial w_1} - \mathbb{E}_e^2[(\phi(X) - Y)^2] \\
&= 4\mathbb{E}_e[(\phi(X) - Y)^3 X_c] - 4\mathbb{E}_e[R(e)] \cdot \mathbb{E}_e[(\phi(X) - Y)X_c].
\end{aligned} \tag{13}$$

Similarly, for $\frac{\partial \operatorname{Var}_e(R(e))}{\partial w_2}$, we have:

$$\frac{\partial \operatorname{Var}_e(R(e))}{\partial w_2} = 4E_e\left[(\phi(X) - Y)^3 \cdot (X_c + n_1 + n_2 + \varepsilon^e)\right] - 4E_e[R(e)] \cdot E_e\left[(\phi(X) - Y) \cdot (X_c + n_1 + n_2 + \epsilon^e)\right]. \tag{14}$$

Now we expand $\phi(X) - Y$:

$$\begin{aligned}
\phi(X) &= w_1 X_c + w_2 X_s \\
X_s &= X_c + n_1 + n_2 + \varepsilon^e \\
\phi(X) &= (w_1 + w_2) X_c + w_2 n_1 + w_2 n_2 + w_2 \varepsilon^e \\
Y &= X_c + n_1 \\
\phi(X) - Y &= (w_1 + w_2 - 1) X_c + (w_2 - 1) n_1 + w_2 n_2 + w_2 \varepsilon^e
\end{aligned} \tag{15}$$

Next, we show that $w_1 = 1, w_2 = 0$ is a optimal solution for minimizing Eqn. 10. We set $w_1 = 1, w_2 = 0$, and plug into $\phi(X)$ and $\phi(X) - Y$, we get:

$$\phi(X) = w_1 X_c + w_2 X_s \rightarrow \phi(X) = 1 \cdot X_c + 0 \cdot X_s = X_c, \tag{16}$$

and

$$\begin{aligned} Y &= X_c + n_1 \\ \phi(X) - Y &= X_c - (X_c + n_1) = -n_1. \end{aligned} \tag{17}$$

Substitute Eqn. 16 17 into Eqn. 13, we get:

$$\begin{aligned} 4\mathbb{E}_e \left[ (\phi(X) - Y)^3 \cdot X_c \right] = 4\mathbb{E}_e \left[ \mathbb{E}_{n_1, n_2} \left[ (-n_1)^3 \cdot X_c \right] \right] = 0 \qquad &(E_e[n_1^3] = 0) \\ E_e[R(e)] = E_e \left[ n_1^2 \right] = \sigma_{n1}^2 & \\ \mathbb{E}_e \left[ \mathbb{E}_{n_1, n_2} \left[ (-n_1) \cdot X_c \right] \right] = 0, & \end{aligned} \tag{18}$$

therefore, we have $\frac{\partial \operatorname{Var}_e(R(e))}{\partial w_1} = 0$. For $\frac{\partial \operatorname{Var}_e(R(e))}{\partial w_2}$ (Eqn. 14), the first term can be expanded as:

$$\begin{aligned} \mathbb{E}_e \left[ (-n_1)^3 \cdot X_c \right] &= 0 \\ \mathbb{E}_e \left[ \mathbb{E}_{n_1, n_2} \left[ (-n_1)^3 \cdot n_1 \right] \right] = -\mathbb{E}_e \left[ \mathbb{E}_{n_1, n_2} \left[ n_1^4 \right] \right] &= -3\sigma_{n1}^4 \\ \mathbb{E}_e \left[ \mathbb{E}_{n_1, n_2} \left[ (-n_1)^3 \cdot n_2 \right] \right] &= 0 \\ \mathbb{E}_e \left[ \mathbb{E}_{n_1, n_2} \left[ (-n_1)^3 \cdot \varepsilon^e \right] \right] &= 0, \end{aligned} \tag{19}$$

thus:

$$4\mathbb{E}_e \left[ \mathbb{E}_{n_1, n_2} \left[ (-n_1)^3 \cdot (X_c + n_1 + n_2 + \varepsilon^e) \right] \right] = 4 \cdot \left( -3\sigma_{n1}^4 \right) = -12\sigma_{n1}^4. \tag{20}$$

For the second term in Eqn. 14:

$$4\mathbb{E}_e[R(e)] \cdot \mathbb{E}_e \left[ \mathbb{E}_{n_1, n_2} \left[ (-n_1) \cdot (X_c + n_1 + n_2 + \varepsilon^e) \right] \right] = 4\sigma_{n1}^2 \cdot \left( -\sigma_{n1}^2 \right) = -4\sigma_{n1}^4, \tag{21}$$

substitute Eqn. 20 and 21 into Eqn. 14, we get:

$$\frac{\partial \operatorname{Var}_e(R(e))}{\partial w_2} = -8\sigma_{n1}^4. \tag{22}$$

As $\sigma_{n1}$ is the standard deviation of $n_1$, where $X_c$ is causally related with $Y$, we assume $\sigma_{n1}$ is a small value, hence the high-order term $\mathcal{O}(\sigma_{n1}^4) \approx 0$. Therefore, we show that $w_1 = 1, w_2 = 0$ is (approxiamately) optimla solution for Eqn. 10, indicating that only the invariant features are used in the classifier $\phi^*(X)$. Now we have:

$$Y = \phi^*(X) = X_c = \sum_i X_{c,i}. \tag{23}$$

Given a fixed value $X_c = x$, there may be multiple combinations for $\{X_{c,i}\}$ that result in $X_c = x$. The set of invariant features $\{X_{c,i}\}$ naturally arises from the learning process of deep neural networks. Some invariant features might be easier to learn due to their salience or higher frequency in the dataset. Consequently, although the VRex objective promotes the learning of invariant encoders (as we have proved above), it does not guarantee the identification of all invariant features $\{X_{c,i}\}$.

For FIIF data generating process, similarly we get:

$$\begin{aligned} \phi(X) &= w_1 X_c + w_2 X_s, \quad X_s = n_2 + \epsilon^e, \quad Y = X_c + n_1, \Rightarrow \\ \phi(X) - Y &= (w_1 - 1)X_c + w_2 n_2 + w_2 \epsilon^e - n_1. \end{aligned} \tag{24}$$

The variance of the loss across environments is:

$$\text{Var}_e\left[(\phi(X)-Y)^2\right] = \mathbb{E}_e\left[(\phi(X)-Y)^4\right] - \left(\mathbb{E}_e\left[(\phi(X)-Y)^2\right]\right)^2. \tag{25}$$

Recall that $\phi(X) = w_1 X_c + w_2 X_s$ and $X_s = n_2 + \epsilon^e$, we have:

$$\phi(X) - Y = (w_1 - 1) X_c + w_2 n_2 + w_2 \epsilon^e - n_1. \tag{26}$$

The squared expected loss term can be expanded as:

$$\begin{aligned}
\mathbb{E}_e\left[(\phi(X)-Y)^2\right] &= \mathbb{E}_e\left[((w_1-1)X_c + w_2 n_2 + w_2 \epsilon^e - n_1)^2\right] \\
&= (w_1-1)^2 \mathbb{E}_e\left[X_c^2\right] + w_2^2 \mathbb{E}_e\left[n_2^2\right] \\
&\quad + w_2^2 \mathbb{E}_e\left[(\epsilon^e)^2\right] + \mathbb{E}_e\left[n_1^2\right] \\
&\quad + 2(w_1-1)w_2 \mathbb{E}_e\left[X_c n_2\right].
\end{aligned} \tag{27}$$

Given the independence between $X_c$, $n_1$, $n_2$ and $\epsilon^e$, we have:

$$\mathbb{E}_e\left[(\phi(X)-Y)^2\right] = (w_1-1)^2 \mathbb{E}_e\left[X_c^2\right] + w_2^2 \mathbb{E}_e\left[n_2^2\right] + w_2^2 \mathbb{E}_e\left[(\epsilon^e)^2\right] + \mathbb{E}_e\left[n_1^2\right] \tag{28}$$

For $\mathbb{E}_e\left[(\phi(X)-Y)^4\right]$, we have:

$$\begin{aligned}
\mathbb{E}_e\left[(\phi(X)-Y)^4\right] &= \mathbb{E}_e\left[((w_1-1)X_c + w_2 n_2 + w_2 \epsilon^e - n_1)^4\right] \\
&= (w_1-1)^4 \mathbb{E}_e\left[X_c^4\right] + w_2^4 \mathbb{E}_e\left[n_2^4\right] \\
&\quad + w_2^4 \mathbb{E}_e\left[(\epsilon^e)^4\right] + \mathbb{E}_e\left[n_1^4\right].
\end{aligned} \tag{29}$$

Taking derivative with respect to $w_1$ and $w_2$:

$$\begin{aligned}
\frac{\partial}{\partial w_1} \mathbb{E}_e\left[(\phi(X)-Y)^2\right] &= 2(w_1-1)\mathbb{E}_e\left[X_c^2\right], \\
\frac{\partial}{\partial w_1} \mathbb{E}_e\left[(\phi(X)-Y)^4\right] &= 4(w_1-1)^3 \mathbb{E}_e\left[X_c^4\right], \\
\frac{\partial}{\partial w_2} \mathbb{E}_e\left[(\phi(X)-Y)^2\right] &= 2w_2\left(\mathbb{E}_e\left[n_2^2\right] + \mathbb{E}_e\left[(\epsilon^e)^2\right]\right), \\
\frac{\partial}{\partial w_2} \mathbb{E}_e\left[(\phi(X)-Y)^4\right] &= 4w_2^3\left(\mathbb{E}_e\left[n_2^4\right] + \mathbb{E}_e\left[(\epsilon^e)^4\right]\right).
\end{aligned} \tag{30}$$

Setting Eqn. 30 to zero, we get $w_1 = 1, w_2 = 0$. In conclusion, for both PIIF and FIIF data generating scenarios, VRex only learns invariant features, however given an additive model, it may only learn a subset of invariant features that are significant or frequently appears in the training set.

$$\square$$

## D.2 THEORETICAL DISCUSSION ON IRMV1 IN LEARNING INVARIANT FEATURES

Using a similar derivation process, we can prove that IRMv1 may only learn a subset of invariant features. We first outline the objective of IRMv1 as follows:

$$\mathcal{L}_{IRMv1} := \min_{w,\phi} \mathbb{E}_e\left[\mathcal{L}(w \circ \phi(X), Y) + \beta \left\|\nabla_{w|w=1.0}\mathcal{L}(w \circ \phi(X), Y)\right\|_2^2\right], \tag{31}$$

here $w$ and $\phi$ denote the classifier and feature extractor, respectively.

**Proposition 4.** *Let* $w^*, \phi^* = \underset{w,\phi}{\text{argmin}} \mathcal{L}_{IRMv1}$, *then* $\widehat{\mathbf{h}}_G = w^*(X)$ *learns invariant features, however* $\widehat{\mathbf{h}}_G$ *may only contain a subset of invariant features, given our data generating assumption:*

$$Y = X_c + n_1, X_s = Y + n_2 + \epsilon^e, \text{ where } X_c := \sum_i X_{c,i}, X_s := \sum_i X_{s,i}, \quad (32)$$

*and FIIF data generating assumption:*

$$Y = X_c + n_1, X_s = n_2 + \epsilon^e, \text{ where } X_c := \sum_i X_{c,i}, X_s := \sum_i X_{s,i}, \quad (33)$$

*when using a linear classifier* $\phi(X) = w_1 X_c + w_2 X_s$, *and* $\mathcal{L}(w \circ \phi(X)) := R(e) = (\phi(X) - Y)^2$.

*Proof.* The IRMv1 objective can be simplied into:

$$\mathcal{L}_{IRMv1} := \left\| \nabla_{w|w=1.0} \mathcal{L}(w \circ \phi(X), Y) \right\|_2^2 = (\phi(X) - Y)^4, \quad (34)$$

given:

$$\begin{aligned} \phi(X) &= w_1 X_c + w_2 X_s \\ X_s &= X_c + n_1 + n_2 + \varepsilon^e \\ Y &= X_c + n_1, \end{aligned} \quad (35)$$

we get:

$$\begin{aligned} \phi(X) &= w_1 X_c + w_2 \left( X_c + n_1 + n_2 + \varepsilon^e \right) = \left( w_1 + w_2 \right) X_c + w_2 n_1 + w_2 n_2 + w_2 \varepsilon^e \\ \phi(X) - Y &= \left( w_1 + w_2 - 1 \right) X_c + \left( w_2 - 1 \right) n_1 + w_2 n_2 + w_2 \varepsilon^\varepsilon. \end{aligned} \quad (36)$$

Taking gradient w.r.t. $w_1$ and $w_2$ respetively:

$$\begin{aligned} \frac{\partial (\phi(X) - Y)^4}{\partial w_1} &= 4(\phi(X) - Y)^3 \cdot \frac{\partial (\phi(X) - Y)}{\partial w_1} \\ &= 4(\phi(X) - Y)^3 \cdot X_c, \end{aligned} \quad (37)$$

and

$$\begin{aligned} \frac{\partial (\phi(X) - Y)^4}{\partial w_2} &= 4(\phi(X) - Y)^3 \cdot \frac{\partial (\phi(X) - Y)}{\partial w_2} \\ &= 4(\phi(X) - Y)^3 \cdot \left( X_c + n_1 + n_2 + \varepsilon^e \right). \end{aligned} \quad (38)$$

Plugging in $w_1 = 1, w_2 = 0$, we get:

$$\begin{aligned} \phi(X) &= X_c \\ \phi(X) - Y &= -n_1, \end{aligned} \quad (39)$$

Substitue Eqn. 39 into Eqn. 37 and Eqn. 38, we get:

$$\mathbb{E}_e \left[ \frac{\partial (\phi(X) - Y)^4}{\partial w_1} \right] = \mathbb{E}_e \left[ \mathbb{E}_{n_1,n_2} \left[ -4n_1^3 \cdot X_c \right] \right] = 0, \quad (40)$$

and

$$\mathbb{E}_e \left[ \frac{\partial(\phi(X) - Y)^4}{\partial w_2} \right] = \mathbb{E}_e \left[ -4n_1^3 \cdot (X_c + n_1 + n_2 + \varepsilon^e) \right]$$
$$= -4\mathbb{E}_e\mathbb{E}_{n_1,n_2} \left[ n_1^3 \cdot X_c \right] - 4\mathbb{E}_e\mathbb{E}_{n_1,n_2} \left[ n_1^3 \cdot n_1 \right] - 4\mathbb{E}_e\mathbb{E}_{n_1,n_2} \left[ n_1^3 \cdot n_2 \right] - 4\mathbb{E}_e\mathbb{E}_{n_1,n_2} \left[ n_1^3 \cdot \varepsilon^e \right],$$
$$\tag{41}$$

given that:

$$\mathbb{E}_{n_1,n_2} \left[ n_1^3 \cdot X_c \right] = 0$$
$$\mathbb{E}_{n_1,n_2} \left[ n_1^3 \cdot n_1 \right] = \mathbb{E}_{n_1,n_2} \left[ n_1^4 \right] = 3\sigma_{n1}^4$$
$$\mathbb{E}_{n_1,n_2} \left[ n_1^3 \cdot n_2 \right] = 0$$
$$\mathbb{E}_{n_1,n_2} \left[ n_1^3 \cdot \varepsilon^e \right] = 0,$$
$$\tag{42}$$

we have $\mathbb{E}_e \left[ \frac{\partial(\phi(X) - Y)^4}{\partial w_2} \right] = -12\sigma_{n1}^4$, with a similar assumption that the standard error $\sigma_{n1}$ of the Gaussian noise $n_1$ of the causal variable $X_c$ is small, we conclude that $\mathbb{E}_e \left[ \frac{\partial(\phi(X) - Y)^4}{\partial w_2} \right] = \mathcal{O}(\sigma_{n1}^4) \approx 0$. Hence IRMv1 learns invariant features under the data generating process. Now we have:

$$Y = \phi^*(X) = X_c = \sum_i X_{c,i}. \tag{43}$$

For FIIF generating process, applying the same technique, we have:

$$\phi(X) - Y = w_1 X_c + w_2 (n_2 + \epsilon^e) - (X_c + n_1) = (w_1 - 1) X_c + w_2 n_2 + w_2 \epsilon^e - n_1.$$
$$\frac{\partial \mathcal{L}_{IRMv1}}{\partial w_1} = 4(\phi(X) - Y)^3 \cdot \frac{\partial}{\partial w_1}(\phi(X) - Y).$$
$$\frac{\partial}{\partial w_1}(\phi(X) - Y) = X_c.$$
$$\tag{44}$$

Therefore:

$$\frac{\partial \mathcal{L}_{IRMv1}}{\partial w_1} = 4(\phi(X) - Y)^3 \cdot X_c = 4 \left( (w_1 - 1) X_c + w_2 n_2 + w_2 \epsilon^e - n_1 \right)^3 X_c.$$
$$\frac{\partial \mathcal{L}_{IRMv1}}{\partial w_2} = 4(\phi(X) - Y)^3 \cdot \frac{\partial}{\partial w_2}(\phi(X) - Y) = 4(\phi(X) - Y)^3 \cdot (n_2 + \epsilon^e)$$
$$\tag{45}$$

Substitute $w_1 = 1, w_2 = 0$ into $\phi(X) - Y$,:

$$\phi(X) - Y = X_c - (X_c + n_1) = -n_1. \tag{46}$$

Taking derivative w.r.t. $w_1$ and $w_2$:

$$\mathbb{E}_e \left[ \frac{\partial \mathcal{L}_{IRMv1}}{\partial w_1} \right] = -4\mathbb{E}_e \left[ n_1^3 X_c \right] = 0, \tag{47}$$

$$\mathbb{E}_e \left[ \frac{\partial \mathcal{L}_{IRMv1}}{\partial w_2} \right] = 4\mathbb{E}_e \left[ n_1^3 (n_2 + \epsilon^e) \right] = 0. \tag{48}$$

Therefore we show that for the widely used FIIF and PIIF data generating process, IRMv1 is able to learn invariant features. However, with a similar reasoning as VRex, there may be multiple combinations of the assignments for $\{X_{c,i}\}$ to lead to a fixed value $X_c = x$, and the objective of IRMv1 (Eqn. 31) won't ensure the learning of all invariant features.

$$\square$$

### D.3 PROOF OF THEOREM 4.1

**Theorem D.1.** *[Restatement of Theorem 4.1] Given that the invariant subgraph $G_c$ contains invariant patterns causally related to the target labels, and $G_s$ contains only spurious patterns, the biased infomax principle achieves spuriosity learning. Specifically, the encoder $h_\theta(\cdot)$ learns solely the spurious features for each data sample in $\mathcal{D}$.*

We begin by stating the assumption underlying our proof, followed by a proof by contradiction.

**Assumption 2.** (*Existence of spuriosity learner*) There exists a learning algorithm capable of achieving spuriosity learning as defined in Def. 2.

Most invariant learning algorithms aim to learn invariant features, and a large collection of literature has demonstrated the existence of such algorithms (Arjovsky et al., 2020; Kreuzer et al., 2021; Parascandolo et al., 2020; Mahajan et al., 2021; Wald et al., 2021; Ahuja et al., 2020; 2021; Chen et al., 2022; Gui et al., 2023; Liu et al., 2022). In contrast, we assume the existence of algorithms that learn spurious features. One such example is provided by Eastwood et al. (2023), where the method learns both invariant and spurious features, supporting the plausibility of our assumption. Recent studies have shown that self-supervised contrastive learning tends to learn spurious features (Yao et al., 2024; Hamidieh et al., 2024; Meehan et al., 2023). Therefore, self-supervised contrastive learning would be one form of the spuriosity learner. Based on the above, we first sketch our proof.

**Proof sketch.** As biased infomax is a more general form of the infomax principle, with exponentially many *node configurations* (Def. 4), the encoder $h_{\theta'}(\cdot)$ under the optimal node configuration must result in a spuriosity learner, which also maximizes the learning objective. We then employ proof by contradiction to demonstrate that the node configuration corresponding to biased infomax in Def 3 leads to a larger mutual information, thereby eliciting a contradiction, leading to the conclusion that the *node configuration* corresponding to Def 3 elicits the spuriosity learner.

**Definition 4.** (*Node configurations*) Let the general form of biased infomax be:

$$\max_\theta \mathbb{E}_{G \sim \mathcal{G}} \frac{1}{|G|} \left( \sum_{v_i \in \widetilde{G}} I\left(\widehat{\mathbf{h}}_i; \widehat{\mathbf{h}}_G\right) - \sum_{v_i \in G \setminus \widetilde{G}} I\left(\widehat{\mathbf{h}}_i; \widehat{\mathbf{h}}_G\right) \right),$$

$$\text{s.t. } \widehat{\mathbf{h}}_i = h_\theta(G), \widehat{\mathbf{h}}_G = \text{READOUT}\left(\widehat{\mathbf{h}}_i\right). \tag{49}$$

A node configuration $\mathbf{c} = \left(c_1, c_2, \ldots, c_{|\mathcal{V}|}\right)^T$, such that $c_i \in \{-1, 1\}$, corresponds to one specific instantiation in Eqn. 49, where $c_i = 1$ denotes that $v_i \in \widetilde{G}$, and $c_i = -1$ means $v_i \in G \setminus \widetilde{G}$. The set of all possible node configurations can be denoted as $\mathcal{H} := \{\mathbf{c} : c_i \in \{-1, 1\}\}$.

Given definition 4, the infomax principle (Eqn. 1) is therefore a special case of the biased infomax principle, where $\mathbf{c} = \mathbf{1}_{|\mathcal{V}|}$. For notation simplicity, let $\mathbf{c} := [\mathbf{c}_1, \mathbf{c}_2]^T, \mathbf{c}_1 \in \mathbb{R}^{|G_s|}, \mathbf{c}_2 \in \mathbb{R}^{|G_c|}$ be two vectors for the node configurations of $G_s$ and $G_c$. Assuming for the optimal node configuration, there are $k$ elements in $\mathbf{c}_1$ with 1, and $|G_s| - k$ elements with $-1$ (denoted as $\mathcal{V}_s^{\mathbf{1}}$ and $\overline{\mathcal{V}_s^{\mathbf{1}}}$), similarly, there are $t$ elements in $\mathbf{c}_2$ with 1, and $|G_c| - t$ elements with $-1$, denoted as $\mathcal{V}_c^{\mathbf{1}}$ and $\overline{\mathcal{V}_c^{\mathbf{1}}}$. We then compare such node configuration $\mathbf{c}^*$ with $\mathbf{c}' = [\mathbf{1}_{|G_s|}, -\mathbf{1}_{|G_c|}]^T$.

*Proof.* First, under the node configuration $\mathbf{c}^*$, the optimal parameter to optimize the learning objective is:

$$\theta^* = \arg\max_\theta \mathbb{E}_{G \sim \mathcal{G}} \frac{1}{|G|} \left( \sum_{v_i \in \mathcal{V}_s^{\mathbf{1}}} I\left(\widehat{\mathbf{h}}_i; \widehat{\mathbf{h}}_G\right) + \sum_{v_i \in \mathcal{V}_c^{\mathbf{1}}} I\left(\widehat{\mathbf{h}}_i; \widehat{\mathbf{h}}_G\right) - \sum_{v_i \in \overline{\mathcal{V}_s^{\mathbf{1}}}} I\left(\widehat{\mathbf{h}}_i; \widehat{\mathbf{h}}_G\right) - \sum_{v_i \in \overline{\mathcal{V}_c^{\mathbf{1}}}} I\left(\widehat{\mathbf{h}}_i; \widehat{\mathbf{h}}_G\right) \right),$$

$$\tag{50}$$

and under the node configuration $\mathbf{c}'$, the optimal parameter is:

$$\theta' = \arg \max_{\theta} \mathbb{E}_{G \sim \mathcal{G}} \frac{1}{|G|} \left( \sum_{v_i \in G_s} I\left(\widehat{\mathbf{h}}_i; \widehat{\mathbf{h}}_G\right) - \sum_{v_i \in G_c} I\left(\widehat{\mathbf{h}}_i; \widehat{\mathbf{h}}_G\right) \right). \tag{51}$$

Let

$$f(\theta^*; \mathcal{D}, \mathbf{c}^*) := \mathbb{E}_{G \sim \mathcal{G}} \frac{1}{|G|} \left( \sum_{v_i \in \mathcal{V}_s^{\mathbf{1}}} I\left(\widehat{\mathbf{h}}_i; \widehat{\mathbf{h}}_G\right) + \sum_{v_i \in \mathcal{V}_c^{\mathbf{1}}} I\left(\widehat{\mathbf{h}}_i; \widehat{\mathbf{h}}_G\right) - \sum_{v_i \in \overline{\mathcal{V}_s^{\mathbf{1}}}} I\left(\widehat{\mathbf{h}}_i; \widehat{\mathbf{h}}_G\right) - \sum_{v_i \in \overline{\mathcal{V}_c^{\mathbf{1}}}} I\left(\widehat{\mathbf{h}}_i; \widehat{\mathbf{h}}_G\right) \right)$$

. Similarly, we can define

$$f(\theta'; \mathcal{D}, \mathbf{c}') := \mathbb{E}_{G \sim \mathcal{G}} \frac{1}{|G|} \left( \sum_{v_i \in G_s} I\left(\widehat{\mathbf{h}}_i; \widehat{\mathbf{h}}_G\right) - \sum_{v_i \in G_c} I\left(\widehat{\mathbf{h}}_i; \widehat{\mathbf{h}}_G\right) \right)$$

.

According to Eqn. 51, we can conclude that $f(\theta'; \mathcal{D}, \mathbf{c}') > f(\theta^*; \mathcal{D}, \mathbf{c}')$, and we also have that $h_{\theta^*}(\cdot) = \widehat{\mathbf{h}}_G$ only learns spurious features as $\theta^*$ is the maximizer for the objective under $\mathbf{c}^*$, and $h_{\theta'}(\cdot) = \widehat{\mathbf{h}}_G$ learns both invariant and spurious features, as $\theta'$ is the maximizer for the objective under $\mathbf{c}'$. Let $\mathbf{h}_G^{\theta^*} = h_{\theta^*}(G)$, and $\mathbf{h}_G^{\theta'} = h_{\theta'}(G)$, therefore we have:

$$I\left(\widehat{\mathbf{h}}_i; \widehat{\mathbf{h}}_G^{\theta^*}\right) = 0, \quad \forall v_i \in G_c; \tag{52}$$

$$I\left(\widehat{\mathbf{h}}_i; \widehat{\mathbf{h}}_G^{\theta'}\right) = c, \quad \forall v_i \in G_c; \tag{53}$$

$$I\left(\widehat{\mathbf{h}}_i; \widehat{\mathbf{h}}_G^{\theta^*}\right) = q, \quad \forall v_i \in G_s; \tag{54}$$

$$I\left(\widehat{\mathbf{h}}_i; \widehat{\mathbf{h}}_G^{\theta'}\right) = q', \quad \forall v_i \in G_s; \tag{55}$$

Eqn. 52 is due to $h_{\theta^*}(\cdot)$ is obtained under $\mathbf{c}^*$, hence only learn spuriosity. Eqn. 53 is due to that $h_{\theta'}(\cdot)$ learns both spurious and invariant features. We also have $q > q'$ due to the same reason. Now we compare $f(\theta'; \mathcal{D}, \mathbf{c}')$ with $f(\theta^*; \mathcal{D}, \mathbf{c}')$ as follows:

$$\begin{aligned} f(\theta^*; \mathcal{D}, \mathbf{c}') - f(\theta'; \mathcal{D}, \mathbf{c}') &= \mathbb{E}_{G \sim \mathcal{G}} \frac{1}{|G|} \left( \sum_{v_i \in G_s} q - \sum_{v_i \in G_c} 0 \right) \\ &\quad - \left( \mathbb{E}_{G \sim \mathcal{G}} \frac{1}{|G|} \left( |G_s| \cdot q' - |G_c| \cdot c \right) \right) \\ &= \mathbb{E}_{G \sim \mathcal{G}} \frac{1}{|G|} \left( |G_s|(q - q') + |G_c| \cdot c \right) > 0, \end{aligned} \tag{56}$$

leading to a contradiction that $f(\theta'; \mathcal{D}, \mathbf{c}') > f(\theta^*; \mathcal{D}, \mathbf{c}')$ for any node configuration $\mathbf{c} \neq \mathbf{c}'$, therefore we conclude that $\mathbf{c}'$ is the optimal configuration that elicits the spuriosity learner, which corresponds to the biased infomax objective in Eqn. 2.

$\square$

## D.4 PROOF OF PROPOSITION 1

**Proposition 5.** *[Restatement of Proposition 1] Given an error rate $p\%$ in the approximation algorithm for $G_c$, let the learning objectives for the biased infomax with the ground-truth subgraphs $G_c$ and $G_s$ and with the approximated subgraphs $\widehat{G}_c$ and $\widehat{G}_s$ be denoted as $\mathcal{L}(\theta^*; \mathcal{D})$ and $\mathcal{L}(\theta'; \mathcal{D})$ respectively. The difference between $\mathcal{L}(\theta^*; \mathcal{D})$ and $\mathcal{L}(\theta'; \mathcal{D})$ can be expressed as:*

$$\mathbb{E}_{G \sim \mathcal{G}} \left[ \frac{2}{|G|} \left( \sum_{v_i \in pG_s} I\left(\widehat{\mathbf{h}}_i; \widehat{\mathbf{h}}_G\right) - \sum_{v_i \in pG_c} I\left(\widehat{\mathbf{h}}_i; \widehat{\mathbf{h}}_G\right) \right) \right].$$

*Proof.* We first expand $\mathcal{L}(\theta^*; \mathcal{D})$ and $\mathcal{L}(\theta'; \mathcal{D})$ as follows.

$$
\mathcal{L}(\theta^*; \mathcal{D}) = \mathbb{E}_{G \sim \mathcal{G}} \frac{1}{|G|} \left( \sum_{v_i \in G_s} I\left(\hat{\mathbf{h}}_i; \hat{\mathbf{h}}_G\right) - \sum_{v_i \in G_c} I\left(\hat{\mathbf{h}}_i; \hat{\mathbf{h}}_G\right) \right)
$$

$$
= \mathbb{E}_{G \sim \mathcal{G}} \frac{1}{|G|} \left( \sum_{v_i \in (1-p)G_s} I\left(\hat{\mathbf{h}}_i; \hat{\mathbf{h}}_G\right) + \sum_{v_i \in pG_s} I\left(\hat{\mathbf{h}}_i; \hat{\mathbf{h}}_G\right) - \sum_{v_i \in (1-p)G_c} I\left(\hat{\mathbf{h}}_i; \hat{\mathbf{h}}_G\right) - \sum_{v_i \in pG_c} I\left(\hat{\mathbf{h}}_i; \hat{\mathbf{h}}_G\right) \right)
$$

and

$$
\mathcal{L}(\theta'; \mathcal{D}) = \mathbb{E}_{G \sim \mathcal{G}} \frac{1}{|G|} \left( \sum_{v_i \in (1-p)G_s \cup pG_c} I\left(\hat{\mathbf{h}}_i; \hat{\mathbf{h}}_G\right) - \sum_{v_i \in (1-p)G_c \cup pG_s} I\left(\hat{\mathbf{h}}_i; \hat{\mathbf{h}}_G\right) \right). \quad (57)
$$

the gap of the two learning objectives $\Delta\mathcal{L}$ is:

$$
\Delta\mathcal{L} := \mathcal{L}(\theta^*; \mathcal{D}) - \mathcal{L}(\theta'; \mathcal{D})
$$

$$
= \mathbb{E}_{G \sim \mathcal{G}} \frac{1}{|G|} \left( \sum_{v_i \in (1-p)G_s} I\left(\hat{\mathbf{h}}_i; \hat{\mathbf{h}}_G\right) + \sum_{v_i \in pG_s} I\left(\hat{\mathbf{h}}_i; \hat{\mathbf{h}}_G\right) - \sum_{v_i \in (1-p)G_c} I\left(\hat{\mathbf{h}}_i; \hat{\mathbf{h}}_G\right) - \sum_{v_i \in pG_c} I\left(\hat{\mathbf{h}}_i; \hat{\mathbf{h}}_G\right) \right)
$$

$$
- \left( \mathbb{E}_{G \sim \mathcal{G}} \frac{1}{|G|} \left( \sum_{v_i \in (1-p)G_s \cup pG_c} I\left(\hat{\mathbf{h}}_i; \hat{\mathbf{h}}_G\right) - \sum_{v_i \in (1-p)G_c \cup pG_s} I\left(\hat{\mathbf{h}}_i; \hat{\mathbf{h}}_G\right) \right) \right).
$$

$$
= \mathbb{E}_{G \sim \mathcal{G}} \frac{2}{|G|} \left[ \sum_{v_i \in pG_s} I\left(\hat{\mathbf{h}}_i; \hat{\mathbf{h}}_G\right) - \sum_{v_i \in pG_c} I\left(\hat{\mathbf{h}}_i; \hat{\mathbf{h}}_G\right) \right].
$$

$$(58)$$

We conclude the proof.

$\square$

### D.5 PROOF OF PROPOSITION 2

**Proposition 6.** *[Restatement of Proposition 2] Given a linear regression model with parameters $\{\theta_1, \theta_2\}$ and spurious features $\{x_1, x_2\}$, the correlation strength for feature $x_1$ is $p$ and for $x_2$ is $1 - p$ when $Y = 0$. Similarly, the correlation strength for feature $x_1$ is $q$ and for $x_2$ is $1 - q$ when $Y = 1$. Assuming the spurious features $x_1$ and $x_2$ can each take values in $\{0, 1\}$, we obtain the following parameter estimates using Mean Squared Error (MSE) loss: $\theta_1 = \frac{q}{p+q}, \theta_2 = \frac{1-q}{2-p-q}$.*

*Proof.* The linear model for predicting $Y$ is: $\hat{Y} = \theta_1 x_1 + \theta_2 x_2$, with Mean Square Error (MSE) loss, for $Y = 0$ with $pn$ samples, $x_1 = 1$ and $x_2 = 0$; with $(1-p)n$ samples, $x_1 = 0, x_2 = 1$; For $Y = 1$ with $qn$ samples, $x_1 = 1, x_2 = 0$, with $(1-q)n$ samples, $x_1 = 0, x_2 = 1$. The MSE loss $\mathcal{L}_{MSE}$ consists of 4 terms:

$$
\mathcal{L}_{MSE} = \sum_{i=1}^{pn} (\theta_1 - 0)^2 + \sum_{i=1}^{(1-p)n} (\theta_2 - 0)^2 + \sum_{i=1}^{qn} (\theta_1 - 1)^2 + \sum_{i=1}^{(1-q)n} (\theta_2 - 1)^2
$$

$$
= pn \cdot \theta_1^2 + (1-p)n \cdot \theta_2^2 + qn \cdot (\theta_1 - 1)^2 + (1-q)n \cdot (\theta_2 - 1)^2
$$

$$
= \frac{1}{4} \left[ p\theta_1^2 + (1-p)\theta_2^2 + q(\theta_1 - 1)^2 + (1-q)(\theta_2 - 1)^2 \right].
$$

Taking partial derivative w.r.t. $\theta_1$:

$$\frac{\partial \mathcal{L}_{MSE}}{\partial \theta_1} = \frac{1}{4} \left[ 2p\theta_1 + 2q\left(\theta_1 - 1\right) \right] = 0$$
$$\Rightarrow p\theta_1 + q\theta_1 - q = 0$$
$$\Rightarrow \theta_1 = \frac{q}{p+q}$$

Similarly for $\theta_2$,

$$\frac{\partial \mathcal{L}_{MSE}}{\partial \theta_2} = \frac{1}{4} \left[ 2(1-p)\theta_2 + 2(1-q)\left(\theta_2 - 1\right) \right] = 0$$
$$\Rightarrow (1-p)\theta_2 + (1-q)\theta_2 - (1-q) = 0$$
$$\Rightarrow (2-p-q)\theta_2 = 1-q$$
$$\Rightarrow \theta_2 = \frac{1-q}{2-p-q}$$

$\square$

As demonstrated in Proposition 2, the presence of spurious feature overlap across different classes hinders the linear classifier's ability to generate distinguishable (symmetric) logits. For instance, when $p = q$, both $\theta_1 = 0.5$ and $\theta_2 = 0.5$, resulting in a loss of distinguishability among different spurious patterns. To mitigate this issue, we propose class-conditioned (intra-class) cross entropy loss which reduces the interference across different classes. To show that class-conditioned (intra-class) cross entropy loss is able to maximize the conditional entropy $H(\mathbf{s}^{(i)} | \widehat{\mathbf{h}}_G^{(i)})$, we propose the following proposition.

**Proposition 7.** *Given the spurious features $\mathbf{h}_s^{(i)}, \forall G^{(i)} \in \mathcal{D}$ and a suitable clustering number $K$ that corresponds to the number of environmental groups in each class $Y = y$, the loss objective $\mathcal{L}_{Inv}$ (Eqn. 3) will maximize the conditional entropy $H(\mathbf{s}^{(i)} | \widehat{\mathbf{h}}_G^{(i)}), \forall i \in \mathcal{D}$.*

To prove that optimizing $\mathcal{L}_{Inv}$ maximizes the conditional entropy $H(\mathbf{s}^{(i)} | \widehat{\mathbf{h}}_G^{(i)})$, we show that $\mathbb{P}(\mathbf{s}^{(i)} | \widehat{\mathbf{h}}_c^{(i)}) \sim Cat(\frac{1}{K})$ with $K$ clusters, hence maximizes $H(\mathbf{s}^{(i)} | \widehat{\mathbf{h}}_G^{(i)})$.

*Proof.* First, we have:

$$\max_\theta H(\mathbf{s}^{(i)} | \widehat{\mathbf{h}}_G^{(i)}) = H(\mathbf{s}^{(i)} | \rho'(\widehat{\mathbf{h}}_G^{(i)})) = H(\mathbf{s}^{(i)} | \widehat{\mathbf{s}}^{(i)}).$$

(59)

Here, $\rho'(\cdot)$ is the classification head for cluster labels $\mathbf{s}^{(i)}$. For notation simplicity, we will omit superscript $(i)$, and denote $\mathbf{s} := \mathbf{s}^{(i)}$ and $\widehat{\mathbf{s}} := \widehat{\mathbf{s}}^{(i)}$. Now using softmax loss, we have:

$$L = -\sum_{j=1}^{K} \mathbf{s}_j \log\left(\sigma(\widehat{\mathbf{s}})_j\right), \text{ where } \sigma(\widehat{\mathbf{s}})_j = \frac{e^{\widehat{\mathbf{s}}_j}}{\sum_{k=1}^{K} e^{\widehat{\mathbf{s}}_k}} \tag{60}$$

$$\frac{\partial \sigma(\widehat{\mathbf{s}})_j}{\partial \widehat{\mathbf{s}}_i} = \sigma(\widehat{\mathbf{s}})_j \left(\delta_{ij} - \sigma(\widehat{\mathbf{s}})_i\right) \tag{61}$$

$$\frac{\partial L}{\partial \widehat{\mathbf{s}}_i} = -\sum_{j=1}^{K} \mathbf{s}_j \frac{\partial \log\left(\sigma(\widehat{\mathbf{s}})_j\right)}{\partial \widehat{\mathbf{s}}_i} \Rightarrow \tag{62}$$

$$\frac{\partial L}{\partial \widehat{\mathbf{s}}_i} = -\left(\mathbf{s}_i - \sigma(\widehat{\mathbf{s}})_i\right). \tag{63}$$

Taking expectation on both side of Eqn. 63, we get:

$$\mathbb{E}\left[\frac{\partial L}{\partial \widehat{\mathbf{s}}_i}\right] = \mathbb{E}\left[-\left(\mathbf{s}_i - \sigma(\widehat{\mathbf{s}})_i\right)\right] \Rightarrow \tag{64}$$

$$0 = -\left(\mathbb{E}\left[\mathbf{s}_i\right] - \mathbb{E}\left[\sigma(\widehat{\mathbf{s}})_i\right]\right) \Rightarrow \tag{65}$$

$$0 = -\left(\frac{1}{K} - \mathbb{E}\left[\sigma(\widehat{\mathbf{s}})_i\right]\right) \Rightarrow \tag{66}$$

$$\mathbb{E}\left[\sigma(\widehat{\mathbf{s}})_i\right] = \frac{1}{K} \tag{67}$$

Eqn. 66 is due to that $\mathbb{E}[\mathbf{s}_i] = \frac{1}{K}$, as each cluster derived from the spurious embedding represents a environmental group, and with appropriate reweighting within each cluster, the samples will be weighted equally across clusters, i.e., samples from majority group and minority group will be weighted equally. Given that this expectation holds over all training samples, one optimal solution for Eqn. 67 is: $\sigma(\widehat{\mathbf{s}})_i = \frac{1}{K}, \forall i \in [K]$ for $K$ class labels. This aligns with the assumption that within each class, there exists a stable pattern $G_c$, and if the encoder is able to effectively capture invariant patterns $G_c$ while discarding spurious correlations, $\sigma(\widehat{\mathbf{s}})_i$ will be a fixed and constant value. With this constraint, the solution $\sigma(\widehat{\mathbf{s}}^{(i)}) = \frac{1}{K}\mathbf{1}_K$ implies that the model learns invariant features. Furthermore, this stable solution maximizes the conditional entropy $H(\mathbf{s}^{(i)}|\widehat{\mathbf{h}}_G^{(i)})$, as $\sigma(\widehat{\mathbf{s}}^{(i)})$ is independent of $\mathbf{s}^{(i)}$.

**Remark.** While $\sigma(\widehat{\mathbf{s}}^{(i)}) = \frac{1}{K}\mathbf{1}_K$ is a sufficient condition to minimize $\mathcal{L}_{\text{inv}}$, there exists other solutions that exploit spurious features to achieve high training accuracy. To guide the model toward the stable solution that satisfies $H(\mathbf{s}^{(i)}|\widehat{\mathbf{h}}_G^{(i)})$, an explicit regularization term, such as $\|\sigma(\widehat{\mathbf{s}}^{(i)}) - \frac{1}{K}\mathbf{1}_K\|_2^2$, can be added to the objective in Eqn. 4. However, our empirical results show that such explicit regularization does not significantly affect performance. This may be attributed to the limited number of class labels and node features in graph-level OOD datasets, which inherently enforce $\sigma(\widehat{\mathbf{s}}^{(i)})$ to be similar across all training samples, facilitating convergence to the stable solution.

$\square$

## D.6 Proof of Theorem 4.2

**Theorem D.2.** *[Restatement of Theorem 4.2] There exists a suitable $\gamma$ and clustering number $K$, such that minimizing the loss objective $\mathcal{L} = \mathcal{L}_{GT} + \lambda\mathcal{L}_{Inv}$ will lead to the optimal encoder $h^*(\cdot)$ which elicits invariant features for any graph $G$, i.e., $\widehat{\mathbf{h}}_G = h^*(G) = \mathbf{h}_c$.*

*Proof.* We aim to show that minimizing the objective $\mathcal{L}$ encourages the encoder $h^*(\cdot)$ to learn representations $\widehat{\mathbf{h}}_G$ that contain only invariant features $\mathbf{h}_c$. First, consider that under ERM, i.e., minimizing $\mathcal{L}_{GT}$ alone, the encoder $h(\cdot)$ learns both invariant and spurious features, given the empirical and theoretical evidence in previous studies (Kirichenko et al., 2023; Chen et al., 2023b). Now, we introduce the invariant loss $\mathcal{L}_{inv}$, according to Proposition 7, will maximize the conditional entropy

$H(\mathbf{s}^{(i)}|\widehat{\mathbf{h}}_G^{(i)})$ for each sample $G^{(i)}$. This encourages the learned representation $\widehat{\mathbf{h}}_G^{(i)}$ to be uninformative about the spurious features $\mathbf{s}^{(i)}$. In the following proof, we drop superscript $i$ for simplicity.

Next, our goal is to show that minimizing $\mathcal{L}$ leads to $\widehat{\mathbf{h}}_G = \mathbf{h}_c$. We consider the following three cases:

*Case 1: The encoder learns only spurious features ($\widehat{\mathbf{h}}_G = \mathbf{h}_s$).* In this case, the representation $\widehat{\mathbf{h}}_G$ is informative about $\mathbf{s}$ but not about $\mathbf{h}_c$. The conditional entropy $H(\mathbf{s}|\widehat{\mathbf{h}}_G)$ is minimized but not maximized.

*Case 2: The encoder learns both invariant and spurious features ($\widehat{\mathbf{h}}_G = \kappa(\mathbf{h}_c, \mathbf{h}_s)$).* Here, $\widehat{\mathbf{h}}_G$ is informative about both $\mathbf{h}_c$ and $\mathbf{h}_s$. While this may minimize $\mathcal{L}_{GT}$ in the training environments, $\widehat{\mathbf{h}}_G$ remains informative about $\mathbf{h}_s$, thus not maximizing $H(\mathbf{s}|\widehat{\mathbf{h}}_G)$.

*Case 3: The encoder learns only invariant features ($\widehat{\mathbf{h}}_G = \mathbf{h}_c$).* In this case, $\widehat{\mathbf{h}}_G$ is uninformative about $\mathbf{h}_s$, therefore, $H(\mathbf{s}|\widehat{\mathbf{h}}_G)$ is greater than previous two cases, furthermore $\widehat{\mathbf{h}}_G$ also minimizes $\mathcal{L}_{GT}$ given the invariant features hold the sufficient predictive power for the targets labels (Assumption 1).

Therefore, we conclude that the encoder $h^*(\cdot)$ will only learn invariant features, as conditioning on spurious features $\mathbf{h}_s$ will not maximize $H(\mathbf{s}|\widehat{\mathbf{h}}_G)$. Additionally, it is important to note that there exists "uninformative" features which may also maximize $H(\mathbf{s}|\widehat{\mathbf{h}}_G)$ without including $\mathbf{h}_c$. This phenomenon has been demonstrated in EQuAD (Yao et al., 2024), where model performance deteriorates when $\lambda > 1$. However, when $\lambda < 1$, $\mathcal{L}_{GT}$ emphasizes useful features that correlate with the target labels, while $\mathcal{L}_{inv}$ acts as a regularization term that encourages the model to remove spurious features and focus on learning invariant features.

$\square$

# E    COMPLEXITY ANALYSIS

We provide time and space complexity analysis for LIRS, followed by empirical running time analysis on GOODMotif-base and OGBG-Molbbbp datasets.

**Space complexity.** The space complexity for LIRS is $\mathcal{O}(|\mathcal{B}|Hm)$, where $|\mathcal{B}|$ denotes the batch size, $H$ denotes the hidden feature dimension, and $m$ denotes the average number of edges. Therefore the memory overhead is on par with ERM, and may outperform other graph data augmentation method such as DIR (Wu et al., 2022c) and GREA (Liu et al., 2022), as for each samples, they generate spurious subgraphs for each sample using $|\mathcal{B}|$ samples in the same minibatch, which leads to $\mathcal{O}\left((|\mathcal{B}|^2 + |\mathcal{B}|)Hm\right)$.

**Time complexity.** The time complexity of LIRS is $\mathcal{O}(kCmF)$, where $k$ is the number of layers in GNN encoder, $C > 1$ is a constant as LIRS runs GNN encoder multiple times, $F$ is the hidden dimension size. Notably, most other graph invariant learning algorithms also exhibit a complexity of $\mathcal{O}(CkmF)$, as they require multiple GNN encoders for subgraph extraction and feature encoding respectively. Therefore, the time cost gap between LIRS and other methods is not significant. We provide a detailed running time analysis using the Motif-base and Ogbg-Molbbbp datasets, as shown in Table 5. On the Motif-base dataset, the biased infomax only requires 20 epochs training, making its time cost comparable to ERM. On the Ogbg-Molbbbp dataset, the time cost of LIRS exceeds that of other methods since the biased infomax requires training for 100 epochs, and GSAT must be run to annotate node labels to enable adaptive biased infomax. We provide a breakdown of the time cost for each stage in LIRS to provide a better understandings in Table 6.

The first three components account for the majority of the time cost in LIRS, however they only need to be run once. Retraining the GNN encoder is both fast and stable, with less variance as $\mathcal{L}_{Inv}$ is merely a cross-entropy loss. This presents a key advantage of LIRS in terms of hyperparameter selection. Specifically, most of OOD methods require hyperparameter search, and for most methods, the process must be restarted entirely for each run, incurring significant time costs. In contrast, for LIRS, only the final GNN retraining step needs to be run multiple times for hyperparameter

Table 5: Running time analysis (in seconds) on various OOD methods

| Method | Motif-base | OGBG-Molbbbp |
|---|---|---|
| ERM | 494.34±117.86 | 92.42±0.42 |
| IRM | 968.94±164.09 | 151.84±7.53 |
| Vrex | 819.94±124.54 | 129.13±12.93 |
| GSAT | 1233±396.19 | 142.47±25.71 |
| GREA | 1612.43±177.36 | 262.47±45.71 |
| CIGA | 1729.14±355.62 | 352.14±93.32 |
| AIA | 1422.34±69.33 | 217.36±11.04 |
| OOD-GCL | 10813±28.12 | 8455.51±68.61 |
| EQuAD | 747.87±34.71 | 278.85±16.64 |
| LIRS | 504.87±24.04 | 421.32±19.86 |

Table 6: Running time analysis (in seconds) of LIRS

| Method | Motif-base | OGBG-Molbbbp |
|---|---|---|
| Biased Infomax | 302.12 | 214.24 |
| MinibatchKmeans+SVM | 6.63 | 3.04±0.35 |
| Mark Nodes | - | 142.47±25.71 |
| GNN Retraining | 196.12±24.04 | 61.57±6.86 |

search, leading to a significantly reduced time cost when conducting multiple runs compared to other methods.

## F    MORE DISCUSSION ON LIRS

**Comparison with EQuAD (Yao et al., 2024).** While LIRS and EQuAD share similarities in the learning paradigm, our study make several distinctions compared with  Yao et al. (2024).  First, while EQuAD primarily addresses the limitations of existing OOD methods that are sensitive to varying spurious correlation strengths, it does not fully explain why the proposed learning paradigm is effective when spurious correlation strength remains stable.  In contrast, our work answer this important question and demonstrates that this learning paradigm enables learning a broader set of invariant features; Second, We curate the *SPMotif-binary* dataset based on the SPMotif datasets, which can serve as a benchmark for future studies to evaluate the effectiveness of methods in learning a broader set of invariant features; Third, While EQuAD uses a standard infomax objective to learn spurious features, we propose a new algorithm that addresses the limitations of the vanilla infomax approach.  Specifically, we introduce the biased infomax to overcome size constraints and further incorporate additional procedure (i.e., a GNN explainer) to annotate critical nodes with adaptive thresholding to realize biased infomax in real-world datasets; Finally, We identify the limitations of the cross-entropy loss in disentangling spurious features and propose a novel loss objective that is more effective in learning invariant features.  Additionally, our proposed loss does not compromise computational efficiency.

**Comparison with Supervised Contrastive Learning (Khosla et al., 2020) and Deep Graph Infomax (Veličković et al., 2018).** The biased infomax principle also share similarity with Supervised Contrastive Learning (SCI) and Deep Graph Infomax (DGI).  However, SCL and DGI primarily target in-distribution (ID) data and are designed to improve performance on predictive tasks.  In contrast, our proposed biased infomax is specifically designed for OOD data, with the goal of learning environment-related features rather than features that directly benefit classification tasks.  This makes our approach conceptually distinct from SCL and DGI; Second, In SCL the contrastive loss operates at the inter-sample level, where explicit labels are available for each image.  However, in our graph-level OOD setting, biased infomax operates at the node level, where such labels are unavailable.  Due to the absence of node labels, we approximate critical nodes using GNN explainer

with adaptive thresholding. This additional procedure also distinct biased infomax from SCL and DGI.

**Comparison with OOD-GCL (Li et al., 2024a).** LIRS and OOD-GCL both utilize contrastive learning to graph invariant features, however, LIRS relies on labeled data in the subsequent stage to effectively learn invariant features, in contrast, OOD-GCL aims to learn invariant features without labeled data, followed by fine-tuning a linear classifier on downstream tasks. This distinction highlights different assumptions and goals in the design of the two methods. Due to the unavailability of labelled data, OOD-GCL underperforms LIRS across all the datasets. In terms of running time, OOD-GCL is also significantly slower than LIRS and all other methods, as it requires multiple rounds of clustering in every epoch and additional invariance regularization, which must be applied for each channel and cluster.

# G    ALGORITHMIC PSEUDOCODE

We provide pseudocode for the proposed LIRS framework, which consists of 3 stages for learning spurious features and invariant features respectively. The code will be made publicly available upon acceptance of our work.

# H    MORE DETAILS ABOUT EXPERIMENTS

## H.1    DATASETS DETAILS

Table 7: Details about the datasets used in our experiments.

| DATASETS | Split | # TRAINING | # VALIDATION | # TESTING | # CLASSES | METRICS |
|---|---|---|---|---|---|---|
| GOOD-HIV | Scaffold | 24682 | 4113 | 4108 | 2 | ROC-AUC |
| | Size | 26169 | 4112 | 3961 | 2 | ROC-AUC |
| GOOD-Motif | Base | 18000 | 3000 | 3000 | 3 | ACC |
| | Size | 18000 | 3000 | 3000 | 3 | ACC |
| OGBG-Molbbbp | Scaffold | 1631 | 204 | 204 | 2 | ROC-AUC |
| | Size | 1633 | 203 | 203 | 2 | ROC-AUC |
| OGBG-Molbace | Scaffold | 1210 | 152 | 151 | 2 | ROC-AUC |
| | Size | 1211 | 151 | 151 | 2 | ROC-AUC |
| SPMotif-binary | Correlation | 6000 | 2000 | 2000 | 2 | ACC |
| SPMotif($\#G_c = 3$) | Correlation | 9000 | 3000 | 3000 | 3 | ACC |

In this subsection, we provide a detailed introduction to the datasets used in this work, the dataset statistics are illustrated in Table 7.

**SPMotif dataset.** Following Wu et al. (2022c), we generate a 3-class synthetic datasets based on BAMotif (Ying et al., 2019). In these datasets, each graph comprises a combination of invariant and spurious subgraphs, denoted by $G_c$ and $G_s$. The spurious subgraphs include three structures (Tree, Ladder, and Wheel), while the invariant subgraphs consist of Cycle, House, and Crane. The task for a model is to determine which one of the three motifs (Cycle, House, and Crane) is present in a graph. A controllable distribution shift can be achieved via a pre-defined parameter $b$. This parameter manipulates the spurious correlation between the spurious subgraph $G_s$ and the ground-truth label $Y$, which depends solely on the invariant subgraph $G_c$. Specifically, given the predefined bias $b$, the probability of a specific motif (e.g., House) and a specific base graph (Tree) will co-occur is $b$ while for the others is $(1 - b)/2$ (e.g., House-Ladder, House-Wheel). When $b = \frac{1}{3}$, the invariant subgraph is equally correlated to the three spurious subgraphs in the dataset. In SPMotif datasets, $S$ is directly influenced by $C$, and $C$ is causally related with $Y$. For the variant of the SPMotif datasets used in Section 3, we attach 3 invariant subgraphs $G_c$ to a base spurious subgraph, and in the test dataset, only 1 invariant subgraph is attached to $G_s$.

**SPMotif-binary dataset.** To evaluate the feature learning quality of various OOD methods, We curate a binary classification dataset based on the SPMotif dataset, which is utilized in Section 5.3.

---

**Algorithm 1** The LIRS framework

---

**Require:** Graph dataset $\mathcal{D}$ with labels $\mathcal{Y}$; training epochs $E$, $E'$; threshold $\tau$; hyperparameter $\gamma$; regularization weight $\lambda$
**Ensure:** Trained GNN model $f^* = \rho \circ h$
1: **Initialize** GNN encoder $h(\cdot)$
2: **for** epoch $t = 1$ to $E$ **do**          ▷ Step 1: Learning spurious features via biased infomax
3:      **for** each graph $G^{(i)}$ in $\mathcal{D}$ **do**
4:          Obtain graph representation $\mathbf{h}_G^{(i)} \leftarrow h(G^{(i)})$
5:          Use GNN explainer $e(\cdot)$ to identify important nodes $\widehat{G}_c^{(i)}$
6:          Remove $\widehat{G}_c^{(i)}$ from $G^{(i)}$ to get $\tilde{G}^{(i)}$
7:          Compute predicted probabilities $\hat{y}^{(i)} \leftarrow \rho(\mathbf{h}_G^{(i)})$ and $\tilde{y}^{(i)} \leftarrow \rho(h(\tilde{G}^{(i)}))$
8:          **if** $|\hat{y}^{(i)} - \tilde{y}^{(i)}| > \tau$ **then**
9:             Obtain $\mathbf{h}_G^{(i)}$ using Eqn. (2) (biased infomax)
10:         **else**
11:             Obtain $\mathbf{h}_G^{(i)}$ using Eqn. (1) (standard infomax)
12:         **end if**
13:      **end for**
14: **end for**
15: Obtain spurious embeddings $\{\mathbf{h}_s^{(i)}\}$ from the trained encoder $h(\cdot)$
16: **for** each class $y$ in $\mathcal{Y}$ **do**          ▷ Step 2: Clustering and Re-fitting Classifier
17:      Collect spurious embeddings $\{\mathbf{h}_s^{(i)} \mid y^{(i)} = y\}$
18:      Perform clustering using Minibatch KMeans on $\{\mathbf{h}_s^{(i)}\}$ to get cluster labels $\{c^{(i)}\}$
19:      Train a linear classifier (e.g., SVM) on $\{\mathbf{h}_s^{(i)}, c^{(i)}\}$ to obtain spurious logits $\mathbf{s}^{(i)}$
20: **end for**
21: Re-initialize GNN encoder $h(\cdot)$, and classification head $\rho(\cdot)$, $\rho'(\cdot)$ ▷ Step 3: Learning invariant features
22: **for** epoch $t = 1$ to $E'$ **do**
23:      **for** each graph $G^{(i)}$ in $\mathcal{G}$ **do**
24:          Obtain graph representation $\widehat{\mathbf{h}}_G^{(i)} \leftarrow h(G^{(i)})$
25:          Compute estimated spurious logits $\widehat{\mathbf{s}}^{(i)} \leftarrow \rho'(\widehat{\mathbf{h}}_G^{(i)})$
26:          Compute reweighting coefficient $w^{(i)} = \dfrac{1 - (\mathbf{s}_j^{(i)})^\gamma}{\gamma}$, where $j$ is the cluster label
27:          Compute invariant loss using Eqn. 3
28:          Update model parameters by minimizing:
29:             $\mathcal{L} = \mathcal{L}_{\text{ERM}} + \lambda \mathcal{L}_{\text{inv}}$
30:      **end for**
31: **end for**
32: **return** Trained model $f^* = \rho \circ h$

---

Specifically, the motifs *House* and *Crane* are assigned label 0, and *Diamond* and *Cycle* are assigned label 1. During the construction of each class's samples, we attached both invariant subgraphs to the base subgraph with $50\%$ chance, while in the remaining $50\%$, we randomly attached one invariant subgraph to the base spurious subgraph. For the test set, we randomly attached a single invariant subgraph to the base subgraph. Similar to the SPMotif dataset, the base spurious subgraph was correlated with the target labels where $b$ controls the correlation strengths, and in the test set an equal correlation strength is assigned for the samples in the same class.

**GOOD-HIV** is a molecular dataset derived from the MoleculeNet Wu et al. (2018) benchmark, where the primary task is to predict the ability of molecules to inhibit HIV replication. The molecular structures are represented as graphs, with nodes as atoms and edges as chemical bonds. Following Gui et al. (2022), We adopt the covariate shift split, which refers to changes in the input distribution between training and testing datasets while maintaining the same conditional distribution of labels given inputs. This setup ensures that the model must generalize to unseen molecular structures that differ in these domain features from those seen during training. We focus on the Bemis-Murcko

scaffold Bemis & Murcko (1996) and the number of nodes in the molecular graph as two domain features to evaluate our method.

**GOOD-Motif** is a synthetic dataset designed to test structure shifts. Each graph in this dataset is created by combining a base graph and a motif, with the motif solely determining the label. The base graph type and the size are selected as domain features to introduce covariate shifts. By generating different base graphs such as wheels, trees, or ladders, the dataset challenges the model's ability to generalize to new graph structures not seen during training. We employ the covariate shift split, where these domain features vary between training and testing datasets, reflecting real-world scenarios where underlying graph structures may change.

**Open Graph Benchmark.** We also use 2 molecule datasets from the graph property prediction task on Open Graph Benchmark Hu et al. (2020) or known as OGBG. They were originally collected by MoleculeNet (Wu et al., 2018) and used to predict the properties of molecules, including (1) blood–brain barrier permeability in MolBBBP; (2) inhibition to human $\beta$-secretase 1 in MolBACE. For all molecule datasets, we use the scaffold splitting procedure as OGBG adopted (Hu et al., 2020). It attempts to separate structurally different molecules into different subsets, which provides a more realistic estimate of model performance in experiments (Wu et al., 2018). In addition, we also adopt size split to evaluate the OOD generalization ability for various OOD methods following Sui et al. (2023); Gui et al. (2022).

## H.2 EXPERIMENTAL SETUP

**Encoding spurious features with biased infomax.** We adopt MVGRL (Hassani & Khasahmadi, 2020) as the contrastive learning method for learning spurious features. To realize instance-level adaptive biased infomax, we first utilize GSAT (Miao et al., 2022) as the GNN explainability framework to identify important nodes in a graph. The biased infomax principle is realized using contrastive learning, with InfoNCE (Oord et al., 2018) loss as the neural mutual information estimator. The estimated nodes from GSAT in a graph $G$ is treated as *negative samples* rather than positive ones. Consequently, for a graph $G$, the graph representation is optimized to align closely with nodes from $G_s$ while diverging from the representation of nodes in $G_c$. The GNN backbone for GSAT is 5-layer GIN (Xu et al., 2018), the hidden dimension is set to 64 for all the datasets except MolHIV, where the hidden dimension is 128. During inference stage, we obtain top-$K$ edges with highest probability in each graph instance and perform counterfactual inference, i.e., after the removal of the subgraph induced by the top-$K$ edges, we record the change in the prediction score $\Delta_{prob}$, and compare with a pre-defined threshold $\tau$ to decide whether it should be biased or not. The nodes in a graph will be treated as negative examples during training MVGRL if $\Delta_{prob} > \tau$. In all the experiments, $K$ is searched over: $\{8, 12\}$, $\tau$ is searched over: $\{0.2, 0.3\}$. For SPMotif datasets, we directly use the ground-truth nodes from $G_c$, and don't employ GSAT for approximation.

**Generating logits from spurious features and target labels.** We use*MiniBatch KMeans* algorithm to obtain the clustering label. To generate (soft) logits as targets from spurious features learned via biased infomax in each cluster, we use linear svm followed by a probability calibrator.

**GNN encoder.** For SPMotif dataset and SPMotif-binary dataset, we adopt 5-layer GIN (Xu et al., 2018) as backbone GNN encoder with mean pooling; For GOOD-Motif datasets, we utilize a 4-layer GIN with sum pooling, and a hidden dimension of 300; For GOOD-HIV datasets, we employ a 4-layer GIN with sum pooling, and a hidden dimension of 128; For the OGBG-Molbbbp and OGBG-Molbace dataset, we adopt a 4-layer GIN with sum pooling, and the dimensions of hidden layers is 64.

**Optimization and evaluation.** By default, we use Adam optimizer (Kingma & Ba, 2014) with a learning rate of $1e-3$ and a batch size of 64 for all experiments. we also employ an early stopping of 10 epochs according to the validation performance for all datasets. Test accuracy or ROC-AUC is obtained according to the best validation performance for all experiments. All experiments are run with 4 different random seeds, the mean and standard deviation are reported using the 4 runs of experiments.

**Baseline setup and hyperparameters.** In our experiments, for the GOOD and OGBG-Molbbbp datasets, the results of ERM, IRM, GroupDRO, and VREx are reported from Gui et al. (2022), while the results for DropEdge, DIR, GSAT, CIGA, GIL, GREA, FLAG, $\mathcal{G}$-Mixup and AIA on

GOOD and OGBG datasets are reported from Sui et al. (2023). To ensure fairness, we adopt the same GIN backbone architecture as reported in Sui et al. (2023). For the remaining datasets and methods, we conduct experiments using the provided source codes from the baseline methods. The hyperparameter search is detailed as follows.

For IRM and VREx, the weight of the penalty loss is searched over $\{1e-1, 1, 1e1, 1e2\}$. The causal subgraph ratio for DIR is searched across $\{1e-2, 1e-1, 0.2, 0.4, 0.6\}$. For RSC, the masking ratio is searched over $\{0.2, 0.3, 0.4\}$. For DiverseModel, the number of classifciation headers is searched over $\{5, 10, 20\}$, and the penalty weight of the diversification loss is searched over $\{1e-1, 1e-2, 1e-3\}$. For DropEdge, the edge masking ratio is seached over: $\{0.1, 0.2, 0.3\}$. For GREA, the weight of the penalty loss is tuned over $\{1e-2, 1e-1, 1.0\}$, and the causal subgraph size ratio is tuned over $\{0.05, 0.1, 0.2, 0.3, 0.5\}$. For GIL, the penalty weight is searched over $\{1e-5, 1e-3, 1e-1, 1.0\}$, and the number of environments is searched over $\{3, 5, 10\}$. For GSAT, the causal graph size ratio is searched over $\{0.3, 0.5, 0.7\}$. For CIGA, the contrastive loss hinge loss weights are searched over $\{0.5, 1.0, 2.0, 4.0, 8.0\}$. For DisC, we search over $q$ in the GCE loss: $\{0.5, 0.7, 0.9\}$. For LiSA, the loss penalty weights are searched over:$\{1, 1e-1, 1e-2, 1e-3\}$. For FLAG, the ascending steps are set to 3 as recommended in the paper, and the step size is searched over $\{1e-3, 1e-2, 1e-1\}$. For AIA, the stable feature ratio is searched over $\{0.1, 0.3, 0.5, 0.7, 0.9\}$, and the adversarial penalty weight is searched over $\{0.01, 0.1, 0.2, 0.5, 1.0, 3.0, 5.0\}$. For EQuAD, the penalty weight is searched over $\{1e-1, 1e-2, 1e-3\}$, and the reweighting coefficient is searched over $\{0.1, 0.3, 0.5, 0.7, 0.9\}$.

**Hyperparameter search for LIRS.** The penalty weight for $\mathcal{L}_{\text{Inv}}$ in LIRS is searched over $\{1e-1, 1e-2, 1e-3\}$. The reweighting coefficient $\gamma$ is searched over $\{0.1, 0.3, 0.5, 0.7, 0.9\}$. The cluster number $C$ is searched over $\{3, 5, 10\}$. The training epoch $E$ at which the spurious embedding is derived from the biased infomax is searched over $\{50, 60, 70, 80, 90\}$ for real-world datasets, and for the synthetic datasets, The training epoch $E$ is searched over $\{5, 6, 7, 8, 9\}$.

**Implementation of LIRS.** We extend the code from GSAT (Miao et al., 2022) to annotate the nodes in each graph to be biased in real-world datasets. We adopt PyGCL (Zhu et al., 2021) package and modify the source code in *DualBranchContrast* to implement the biased infomax to generate spurious embeddings. To generate logits from the spurious embeddings, we use *MiniBatchKMeans*, *linearSVC*, and *CalibratedClassifierCV* in Scikit-Learn package (Pedregosa et al., 2011). We use PyTorch Paszke et al. (2019) and Pytorch-Geometric Fey & Lenssen (2019) for the remaining implementation.

## H.3    ADDITIONAL EXPERIMENTAL RESULTS

**Visualization of latent features derived from the biased infomax.** We provide visualization of the 2d latent embedding derived form the biased infomax. Specifically, we utilize the SPMotif datasets and GOODMotif datasets to assess the correlation between the learned embeddings after dimensionality reduction and environment labels within each class. We annotate each data sample with the environment label corresponding to three spurious patterns (Tree, Ladder, and Wheel). We then use the spurious features obtained from the biased infomax objective, apply t-SNE Van der Maaten & Hinton (2008) for dimensionality reduction, and use KMeans for clustering and visualization. The epoch at which the latent embedding is obtained is selected according to the best hyperparameter. In Figure 6, points of different colors correspond to different environment labels. It can be observed that within each class, the clusters highly correlate with environments, indicating that the biased infomax effectively captures different spurious patterns.

## H.4    SOFTWARE AND HARDWARE

All the experiments are ran with PyTorch (Paszke et al., 2019) and PyTorch Geometric (Fey & Lenssen, 2019). We run all the experiments on Linux servers with RTX 4090 and CUDA 12.2.

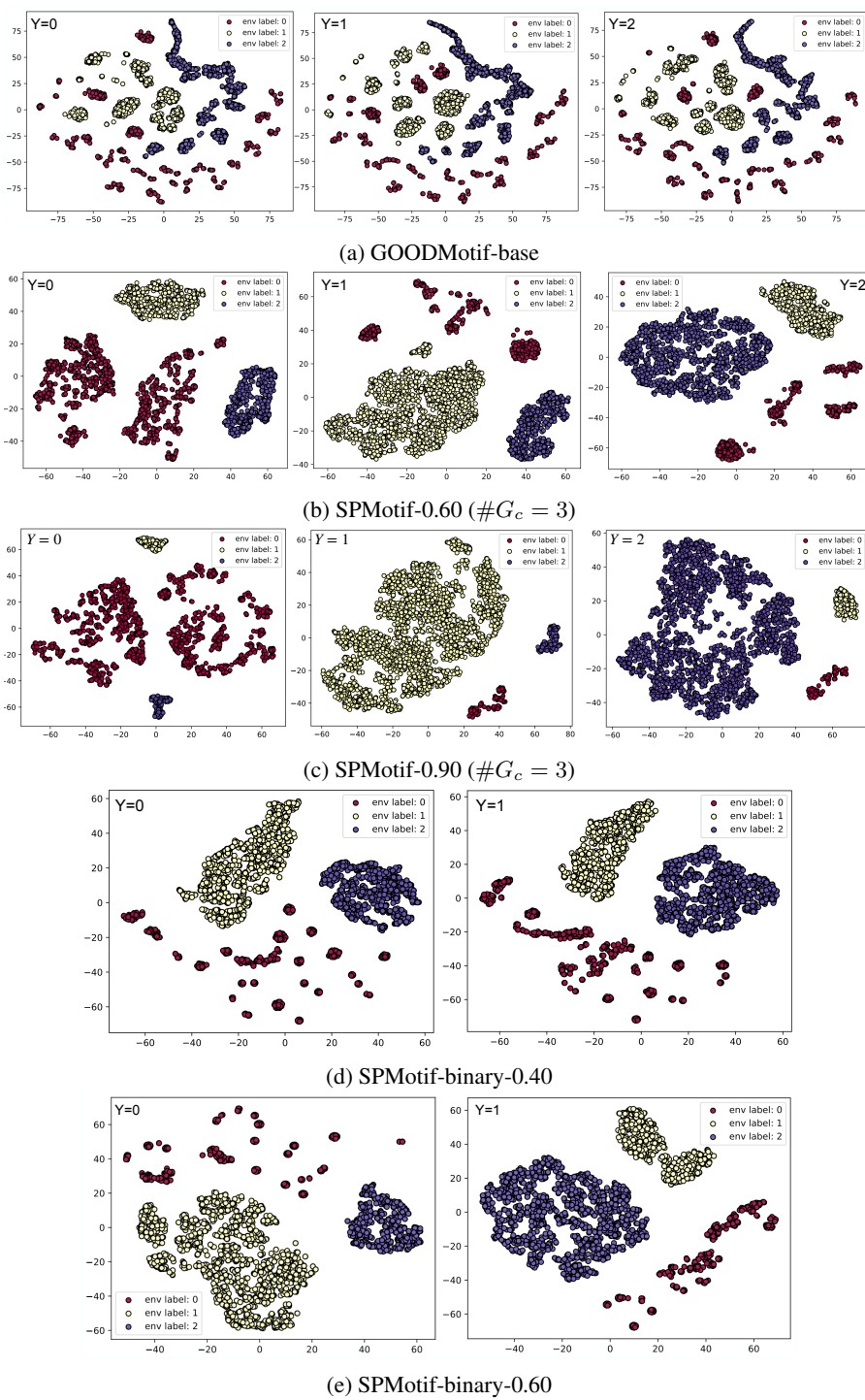

Figure 6: Clustering visualizations using latent embeddings derived from the biased infomax. The clusters resulted from latent embeddings obtained from the biased infomax are highly correlated with the environment labels.

