# OpenReview forum: "Learning Graph Invariance by Harnessing Spuriosity"
_ICLR.cc/2025/Conference — ICLR 2025 Poster_

### Official Review · Reviewer_vWa5 · 2024-10-29

**Soundness:** 2
**Presentation:** 3
**Contribution:** 2
**Rating:** 6
**Confidence:** 3

**Summary:**

This paper presents a graph invariant learning method based on spurious feature learning, termed LIRS. The authors argue that directly learning invariant features is challenging, prompting them to first learn spurious features in order to decouple them from graph representation to get the invariant parts, thus this approach necessitates training the encoder for two times, separately. They introduce a learning objective called bias infomax for the spurious feature encoder, and a loss function $L_{inv}$ designed to encourage the encoder to learn invariant features. The authors validate the effectiveness of both loss functions through theoretical analysis.


However, I find the theoretical feasibility of $L_{inv}$  to be somewhat questionable (see the question part). Additionally, the proposed method appears to be more complex than standard graph invariant learning methods, yet there is no discussion of its efficiency (e.g., time complexity or reported training times), which raises concerns for me. Therefore, I rated it a 5, anticipating that the authors will address my issues, which may prompt me to reconsider my score.

**Strengths:**

1. I agree that learning invariant features indirectly is intuitively likely to be easier and achieve better performance than a direct approach.

2. The authors provide some experimental analysis to support its motivation, as well as some theoretical analysis to justify the two optimization objectives.

3. Good experimental performance.

**Weaknesses:**

1. The author lacks a clear description of the differences between this method and the existing methods. As far as I know, for example, EQuAD, which the author has cited many times, I checked the article and found that the model pipeline of LIRS is very close to it, and the differences seem to be only whether utilize bias infomax and the definition of $s$ in $L_{inv}$.

2. Missing details about the model, such as how to optimize bias infomax? As far as I know mutual information is hard to calculate, how does GSAT get the subgraph? Why choose this approach?

**Questions:**

1.  About $L_{inv}$, I think it's implemented using cross entropy, $\hat{s}^{i}$ is the prediction of the environmental label ${s}^{(i)}$ by using $\hat{h_G}$ , and look forward to by minimizing the cross entropy to make prediction closer to real label. Intuitively, i don't think this is helpful for learning invariant learning, because if the learned representation is the invariant feature, I don't think $\hat{h_G}$ should be able to predict the correct  ${s}^{(i)}$, since it should only be related to labels. Mathematically speaking, the authors claim in Proposition 7 in the Appendix that minimizing $L_{inv}$ is equivalent to maximizing the conditional entropy $H(s^{(i)}|\hat{h_G^i})$,
which I doubt is valid. Known $L_ {inv} $ is cross entropy, we have a $H(s^{(i)}|\hat{h_G^i})$ = $H(s^{(i)},\hat{h_G^i})$-$H(\hat{h_G^i})$ .Minimizing the cross-entropy loss ($H(s^{(i)},\hat{h_G^i})$) fails to maximize the conditional entropy.
2. LIRS requires multi-step training and involves multiple models (GSAT,SVM, etc.). Is it less efficient than existing graph-invariant learning models?
3. What is the difference between you and the existing method (EQuAD)? Is LIRS an incremental improvement version of it?

---

> ### Author Response · Authors · 2024-11-20
> **Response to Reviewer vWa5 (Part 1/3)**
>
> *We sincerely thank you for your valuable feedback and constructive suggestions in reviewing our paper! Please see below for our responses to your comments and concerns.*
>
> ---
>
> >**W1: Compare and contrast with EQuAD**
>
> **A1:** Thank you for the thoughtful question.
>
> We agree with the reviewer that the model pipeline of LIRS shares similarities with EQuAD. However, our work makes several distinct contributions compared to EQuAD, including:
>
> - **Motivation and new insights:** While EQuAD primarily addresses the limitations of existing OOD methods that are sensitive to varying spurious correlation strengths, it does not fully explain why the proposed learning paradigm is effective when spurious correlation strength remains stable. In contrast, **our work answer this important question and demonstrates that this learning paradigm enables learning a broader set of invariant features.**
> - **New Dataset Curation:** We curate the SPMotif-binary dataset based on the SPMotif datasets, which can **serve as a benchmark for future studies to evaluate the effectiveness of methods in learning a broader set of invariant features.**
> - **New Algorithm:** While EQuAD uses a standard infomax objective to learn spurious features, we propose a new algorithm that addresses the limitations of the vanilla infomax approach. Specifically, we introduce _the biased infomax_ to **overcome size constraints and further incorporate additional procedure to annotate critical nodes with adaptive thresholding to realize biased infomax in real-world datasets**.
> - **New Disentanglement Loss:** We **identify the limitations** of the cross-entropy loss in disentangling spurious features and propose a novel loss objective that is **more effective in learning invariant features**. Additionally, our proposed loss **does not compromise computational efficiency.**
> - In our work, $s$ refers to class-specific cluster-based targets, which differ from $s$ in EQuAD, where SVM-derived logits are used as targets. Our theoretical and experimental analyses demonstrate that **the proposed loss objective enables more effective learning of invariant features without hurting computational efficiency.**
>
> >**W2.1: How to optimize biased infomax**
>
> **A2:** We utilize the PyGCL package for the implementation of biased infomax. Specifically, in the optimization of vanilla deep graph infomax, the representation $\hat{h}\_G$ of each graph $G$ uses nodes from other graphs as negative samples, while nodes belonging to $G$ serve as positive samples.
>
> - In biased infomax, we first annotate critical nodes that are likely to belong to $G_c$ and **designate them as negative samples**. Consequently, for any graph $G$, the graph representation is designed to **align closely with nodes from $G_s$ while diverging from the representation of nodes in $G_c$**, effectively learning the spurious features.
> - For some graphs the learning objective will be reverted back to vanilla infomax when the thresholding condition is not satisfied.
> - In summary, **the biased infomax principle can be easily implemented using contrastive learning from the off-the-shelf packages.**
>
> >**W2.2: How does GSAT get the subgraph, and why use it**
>
> **A3:**
>
> - Similar to other OOD methods, GSAT employs a **subgraph selector** along with the Information Bottleneck principle as regularizer to train the model. By leveraging the sampling probability of each edge derived by the trained subgraph selector, we can identify the critical nodes that are most likely to belong to $G_c$.
> - The primary reason for choosing GSAT is its simplicity and robustness. **It does not require tuning hyperparameters and is resilient to distribution shifts**, striking a balance between efficiency and effectiveness. In contrast, other OOD methods often involve many hyperparameters, which could increase the complexity of LIRS.
> - However, LIRS is a **flexible framework** and can be adapted to use other methods to identify critical nodes, which may lead to enhanced OOD performance.

---

> ### Author Response · Authors · 2024-11-20
> **Response to Reviewer vWa5 (Part 2/3)**
>
> >**Q1: How does $\mathcal{L}\_{inv}$ work?**
>
> **A4:** Thank you for raising this point. We believe there might be some potential misunderstandings regarding how $\mathcal{L}_{inv}$ works in LIRS.
> - To clarify, $\mathcal{L}\_{GT}$ uses the ground-truth labels as targets, while **$\mathcal{L}\_{inv}$  employs $s\_j^{(i)}$ as the target**, although also adopts cross-entropy loss. Here, **$s\_j^{(i)}$ is obtained from biased infomax**, which effectively captures spurious patterns. By performing clustering, we derive the cluster labels and treat them as environment labels. The estimated logits, $\hat{s}\_j^{(i)} := \rho'(h(x))$, are produced by the GNN encoder followed by a specific linear layer designed to estimate spurious logits. By using $s\_j^{(i)}$ as the target labels, the spurious features identified through biased infomax are incorporated into the learning process.
> - An intuitive example illustrates how $\mathcal{L}\_{inv}$ maximizes the conditional entropy $H(s_j^{(i)} | \hat{s}_j^{(i)})$:
>     - __Example.__ Suppose there are 100 samples in each cluster for $Y=0$, and for simplicity, we assume two clusters. The cluster labels are $[0, 1]^T$ and $[1, 0]^T$ for the two clusters, respectively. By minimizing the cross-entropy loss, since both clusters have equal sample sizes, the model will output $\hat{s}\_j^{(i)} = [0.5, 0.5]^T$ for maximal randomness. This output provides no information about $s\_j^{(i)}$, hence maximizing the conditional entropy $H(s\_j^{(i)} | \hat{s}\_j^{(i)})$. As it requires the sample size or weights to be equal, we adopt GCE function to upweight the minority group to satisfy this condition.
>     - Building on this, **$\mathcal{L}\_{inv}$ regularizes the model to learn only invariant features**. Since if the model learns spurious features, the conditional entropy would not be maximized. This ensures that **the learned features are indeed invariant features in the latent space.**
>
> >**Q2: Efficiency of LIRS**
>
> **A5:** Thank you for raising this point.
>
> - The time complexity of LIRS depends on the specific GNN encoder employed. In this work, we use GIN, a 1-WL GNN encoder. The time complexity is **$\mathcal{O}(CkmF)$**, where $k$ is the number of GNN layers, $m$ is the average number of edges, and $F$ is the hidden feature dimension. LIRS introduces an additional constant factor $C > 1$ due to its 2-stage paradigm. Notably, **most other graph invariant learning algorithms also exhibit a complexity of $\mathcal{O}(CkmF)$**, as they require multiple GNN encoders for subgraph extraction and feature encoding respectively. **Therefore, the time cost gap between LIRS and other methods is not significant.**
> -  **We have added a detailed time analysis using the Motif-base and Ogbg-Molbbbp datasets, and revised our draft in Appendix E, highlighted in $\textcolor{blue}{\text{blue}}$.**
>
> **Table 1: Run time analysis on the Motif-base and Ogbg-Molbbbp datasets.**
> | Method | GOODMotif-base | OGBG-Molbbbp |
> |:---:|:---:|:---:|
> | ERM | 494.34±117.86 | 92.42±0.42 |
> | IRM | 968.94±164.09 | 151.84±7.53 |
> | Vrex | 819.94±124.54 | 129.13±12.93 |
> | GSAT | 1233±396.19 | 142.47±25.71 |
> | GREA | 1612.43±177.36 | 262.47±45.71 |
> | CIGA | 1729.14±355.62 | 352.14±93.32 |
> | AIA | 1422.34±69.33 | 217.36±11.04 |
> | EQuAD | 747.87±34.71 | 278.85±16.64 |
> | LIRS (Ours) | 504.87±24.04 | 421.32±19.86 |
>
> - On the Motif-base dataset, the biased infomax only requires 20 epochs training, making its time cost comparable to ERM.
> - On the Ogbg-Molbbbp dataset, the time cost of LIRS exceeds that of other methods since the biased infomax requires training for 100 epochs, and GSAT must be run to annotate node labels to enable adaptive biased infomax.
> - Below, **we provide a breakdown of the time cost for each stage:**
>
> **Table 2: Break down of running time for each stage in LIRS.**
> | Method | GOODMotif-base | OGBG-Molbbbp |
> |:---:|:---:|:---:|
> | Biased Infomax | 302.12 | 214.24 |
> | MinibatchKmeans+SVM | 6.63 | 3.04±0.35 |
> | Mark Nodes | - | 142.47±25.71 |
> | GNN Retraining | 196.12±24.04 | 61.57±6.86 |
>
> - The first three components account for the majority of the time cost; however, __they only need to be run once__. Retraining the GNN encoder is both **fast and stable**, with less variance as $\mathcal{L}\_{Inv}$ is merely a cross-entropy loss.
> - __This presents a key advantage of LIRS in terms of hyperparameter selection__. Specifically, most of OOD methods require hyperparameter search, and for most methods, **the process must be restarted entirely for each run, incurring significant time costs.** In contrast, for LIRS, **only the final GNN retraining step needs to be run multiple times for hyperparameter search, leading to a significantly reduced time cost when conducting multiple runs compared to other methods.**

---

> ### Author Response · Authors · 2024-11-20
> **Response to Reviewer vWa5 (Part 3/3)**
>
> >**Q3: Is LIRS an improved version of EQuAD**
>
> **A6:** Despite LIRS and EQuAD adopt the same learning paradigm, **LIRS introduces significant technical contributions to this paradigm and addresses an important research question: _What is the advantage of this new learning paradigm?_** In this process, we also introduce **new datasets curation** specifically designed to evaluate the extent of invariant features captured by different OOD methods. These contributions highlight the novelty of LIRS, therefore we respectfully claim that our work is not merely an incremental improvement over EQuAD.
>
> ---
>
> We sincerely thank you for your careful review and insightful comments, and we hope we have addressed your concerns on the soundness, efficiency and novelty of our approach.

---

> ### Comment · Reviewer_vWa5 · 2024-11-21
> **Response**
>
> Thanks for your response！ I understand and agree with your motivation: identify the spurious feature first and remove it from the original representation. My question is that since $L_{inv}$ is a cross-entropy loss, minimizing it is, by definition, equivalent to minimizing the Joint entropy $H(\hat{s},s)$. However, $H(s|\hat {s}) $ equals $H(\hat{s},s) - H(\hat {s})$, then minimize the cross entropy ($H (\hat{s}, s) $) equal to minimize condition entropy, not maximize it! Thus I think you should amend the final loss function as $L_ {GT} - L_ {inv} $, In order to ensure maximum $H (\hat{s}, s) $ so as to maximize the $H (s | \hat {s}) $.
>
> This is my main confusion and why I still rate  5 for now.

---

> ### Author Response · Authors · 2024-11-21
> **Addressing the concern of $\mathcal{L}\_{inv}$**
>
> Thank you for engaging with our work and for the follow-up question.
>
> ---
>
> The gap and misunderstandings may arise from **the relationship between the cross-entropy loss and the joint Shannon entropy**. Let us clarify this by presenting their definitions:
>
>    (1) **Cross-entropy:**
>    $$
>    H_q(p) = \sum_x p(x) \log \left(\frac{1}{q(x)}\right).
>    $$
>
>    Cross-entropy measures the **difference between two probability distributions**, $p(\cdot)$ (the target distribution) and $q(\cdot)$ (the predicted distribution).
>
>    (2) **Joint entropy:**
>    $$
>    H(x, y) = \sum_{x, y} p(x, y) \log \left(\frac{1}{p(x, y)}\right).
>    $$
>
>    Joint entropy measures **the uncertainty** in the joint distribution $p(x, y)$.
>
>    Therefore, we respectfully emphasize that **cross-entropy and joint entropy are not equivalent, and minimizing cross-entropy does not imply minimizing joint entropy.**
>
> Next, we wish to provide a clearer explanation of why minimizing $\mathcal{L}\_{inv}$ effectively maximizes the conditional entropy $H(s \mid \hat{s})$.
>
>    - Consider $s\_j^{(i)}$ as the targets and $x^{(i)}$ as the input (i.e., a graph sample). Let $\hat{s}\_j^{(i)} := \rho'(h(x^{(i)}))$, where $\rho'(\cdot)$ is the linear classifier. Minimizing $\mathcal{L}\_{inv}$ corresponds to minimizing the cross-entropy between $\hat{s}$ and $s$, thereby aligning the distribution of $\hat{s}$ with $s$.
>
>    - As illustrated in the example above, if the targets are **equally** split into $[0, 1]^T$ and $[1, 0]^T$ (via sample reweighting), **the optimal solution is $\hat{s}_j^{(i)} = [0.5, 0.5]^T$ for all samples**. At this point, the model parameters $\theta^* := \rho^{\prime*} \circ h^*(\cdot)$ converge such that **the predicted logits $\hat{s}_j^{(i)}$ are independent of the samples.**
>
>    - This implies that **$\hat{s}$ is independent of $s$, i.e., $\hat{s} \perp s$.**
>
>    - Using this independence property, we conclude that **$H(s \mid \hat{s})$ is maximized**. Specifically, when $\hat{s} \perp s$, $H(s \mid \hat{s}) = H(s)$, which reaches the upper bound of the conditional entropy.
>
>
> ---
>
> We hope our new response can address your concerns regarding how $\mathcal{L}\_{inv}$ works. We will include the example and more discussions on why $\mathcal{L}\_{inv}$ maximizes the conditional entropy in our revised draft. We are happy to engage further if there are any other questions!

---

> > ### Comment · Reviewer_vWa5 · 2024-11-21
> > **Response**
> >
> > Thanks for your reply! Sorry for my confusion between joint entropy and cross-entropy, but i still have question about, by optimizing $L_{inv}$,why the the optimal solution is [0.5,0.5] but not a one hot vector?

---

> > > ### Author Response · Authors · 2024-11-21
> > > **Reply to Reviewer vWa5**
> > >
> > > Thank you for your quick reply!
> > >
> > > **Why the optimal solution is $[0.5, 0.5]^T$**
> > >
> > > To address your concern, let us use a binary classification case to demonstrate that **$s^{(i)} = \frac{1}{2}$ is an optimal point.**
> > >
> > > - For binary classification with labels $s \in \\{0, 1\\}$ and predictions $\hat{s}$, the loss $\mathcal{L}\_{\text{inv}}$ is defined as:
> > >
> > > $$
> > > \mathcal{L}\_{\text{inv}} = -\frac{1}{n} \sum_{i=1}^n \left[s^{(i)} \log \hat{s}^{(i)} + (1 - s^{(i)}) \log (1 - \hat{s}^{(i)})\right].
> > > $$
> > >
> > > Assuming equal sample sizes for each class, i.e., $n_1 = n_2 = \frac{n}{2}$, the derivative of the loss with respect to $\hat{s}$ is:
> > >
> > > $$
> > > \frac{\partial \mathcal{L}\_{\text{inv}}}{\partial \hat{s}} = -\frac{1}{n} \left[\sum\_{i \in \text{class } 1} \frac{1}{\hat{s}^{(i)}} - \sum\_{i \in \text{class } 0} \frac{1}{1 - \hat{s}^{(i)}}\right].
> > > $$
> > >
> > > At the optimal point, the gradient should **equal zero**, i.e.,
> > >
> > > $$
> > > \sum_{i \in \text{class } 1} \frac{1}{\hat{s}^{(i)}} = \sum_{i \in \text{class } 0} \frac{1}{1 - \hat{s}^{(i)}}.
> > > $$
> > >
> > > Given that $n_1 = n_2 = \frac{n}{2}$, the optimal solution is $\hat{s}^{(i)} = \frac{1}{2}$. Substituting $\hat{s}^{(i)} = \frac{1}{2}$ into the derivative shows that:
> > >
> > > $$
> > > \frac{\partial \mathcal{L}_{\text{inv}}}{\partial \hat{s}} = 0.
> > > $$
> > >
> > > Thus, $\hat{s}^{(i)} = \frac{1}{2}$ for all samples $i$ is the optimal solution.
> > >
> > > ---
> > >
> > > We hope this response has addressed your concern. Please feel free to reach out if you have any further questions.

---

> > > > ### Comment · Reviewer_vWa5 · 2024-11-21
> > > > **Response**
> > > >
> > > > Thanks for your response! I understand what you mean, maybe my knowledge level is limited, but I think your conclusion is very strange, **why optimize a loss function about classification to reach the optimal, the corresponding prediction probability of the correct class is equal to the probability of random classification （1/2 for binary classification ）**, I'm sorry that I can't understand, maybe I missed something?
> > > >
> > > > Also, regarding the formula in your appendix on the proof of Proposition7, I am curious why Eq. 64 takes expectation operation? And why Eq. 67 can be translated to 68, that is, **the value of a random variable is equal to its expectation**.
> > > >
> > > > I sincerely look forward to the author's answers, because I hope to give a fair score with the details of the paper being clarified!

---

> ### Author Response · Authors · 2024-11-21
> **More discussions on the invariance regularization (Part 1/2)**
>
> We sincerely thank the reviewer for raising a **valuable and important** point for understanding why and how $\mathcal{L}_{inv}$ can work. Below, we address each concern in detail.
>
> ## **Why $\mathcal{L}_{\text{inv}}$ leads to $\hat{s}^{(i)} = \frac{1}{2}$**
>
> - As discussed in our previous response, $\hat{s}^{(i)} = \frac{1}{2}$ is **an optimal point** for minimizing $\mathcal{L}_{\text{inv}}$. This solution implies that the learned features $h(x)$ are independent of the spurious features (i.e., $s$). Consequently, **the learned representation relies solely on invariant features, effectively ignoring spurious correlations.**
>
> - This behavior aligns with our assumption that, for each class, there exists a stable pattern $G_c$. This corresponds to **adding a constraint** to:
>
> $$
> \sum_{i \in \text{class } 1} \frac{1}{\hat{s}^{(i)}} = \sum_{i \in \text{class } 0} \frac{1}{1 - \hat{s}^{(i)}}, s.t. \hat{s}^{(i)} = c.
> $$
>
> - Since within each class, the encoder captures $G_c$ (the invariant features), therefore $\hat{s}^{(i)}$ must be equal to a fixed value. **This explains why random classification with equal probability aids in learning invariant features.**
>
> ## **Addressing Degrees of Freedom in the Optimality Condition**
>
> - The reviewer’s **major concerns** likely arise from the observation that the constraint:
>
> $$
> \sum_{i \in \text{class } 1} \frac{1}{\hat{s}^{(i)}} = \sum_{i \in \text{class } 0} \frac{1}{1 - \hat{s}^{(i)}},
> $$
>
> **allows for many degrees of freedom**. As a result, solutions other than $\hat{s}^{(i)} = \frac{1}{2}$ may also minimize $\mathcal{L}_{\text{inv}}$. These solutions can achieve **high classification accuracy by exploiting spurious correlations**. To illustrate, consider a binary classification case with 3 samples per class:
>
> $$
> \frac{1}{\hat{s}^{(1)}} + \frac{1}{\hat{s}^{(2)}} + \frac{1}{\hat{s}^{(3)}} = \frac{1}{1 - \hat{s}^{(4)}} + \frac{1}{1 - \hat{s}^{(5)}} + \frac{1}{1 - \hat{s}^{(6)}}.
> $$
>
> While $\hat{s}^{(i)} = \frac{1}{2}$ is a valid solution, another solution could be:
>
> $$
> \hat{s}^{(1)} = 0.6, \hat{s}^{(2)} = 0.7, \hat{s}^{(3)} = 0.8,
> $$
> and
> $$
> \hat{s}^{(4)} = 0.4, \hat{s}^{(5)} = 0.3, \hat{s}^{(6)} = 0.2.
> $$
>
> This solution achieves **100% classification accuracy on the training set with threshold is 0.5** by exploiting spurious correlations. Similarly, other solutions could achieve high accuracy without relying on invariant features, instead **depending on spurious correlations.**
>
> ## **How to select $\hat{s}^{(i)} = \frac{1}{2}$ as the desired solution**
>
> This raises a question: **How to select the stable solution that all $\hat{s}^{(i)}=\frac{1}{2}$?**
>
> To ensure that the solution $\hat{s}^{(i)} = \frac{1}{2}$ is selected, we adopt a method similar to other OOD methods: using an **OOD validation set** for model selection via cross-validation. The OOD validation set introduces **distribution changes in spurious features**, causing solutions that rely on spurious correlations to **underperform the stable solution.**
>
> **By selecting the model with the highest validation performance, we effectively choose the model parameters where $\hat{s}^{(i)} = \frac{1}{2}$ for all $i$**. This stable solution ensures that the model learns invariant features.
>
> ## **Why does training the model using $\mathcal{L} = \mathcal{L}\_{GT} + \lambda \mathcal{L}\_{inv}$ lead to convergence to a stable solution?**
>
> We believe this is the final question that is **central to the reviewer's concern**, which we address as follows:
>
> - In graph-level OOD datasets, **node features and number of class labels are quite limited**. As a result, $\hat{s}^{(i)}$ **does not diverge significantly across different $i$**, implicitly enforcing the condition:
>
>    $$
>    \sum_{i \in \text{class } 1} \frac{1}{\hat{s}^{(i)}} = \sum_{i \in \text{class } 0} \frac{1}{1 - \hat{s}^{(i)}}, s.t. \hat{s}^{(i)}=c.
>    $$
>
> This property facilitates optimization and drives $\hat{s}^{(i)}$ to converge to $\frac{1}{2}$.
> - Since the invariant features are sufficiently predictive to the targets,  the stable solution **does not** hurt the performance in the training set, therefore **$\mathcal{L}$ can reach to optimal solution, meanwhile $\hat{s}^{(i)}=\frac{1}{2}$ is also satisfied**.
> - Earlier we experimented with adding an explicit regularization term to enforce $\hat{s}^{(i)}-\frac{1}{2} = 0$, **the performance was similar to using $\mathcal{L} = \mathcal{L}\_{GT} + \lambda \mathcal{L}\_{inv}$ alone**.
> - This suggests that the inherent properties of the dataset and the combined loss function are **sufficient to guide the model toward a stable solution.**

---

> ### Author Response · Authors · 2024-11-21
> **More discussions on the invariance regularization (Part 2/2)**
>
> In summary:
> - $\hat{s}^{(i)} = \frac{1}{2}$ is **one of the optimal solutions** for $\mathcal{L}_{\text{inv}}$, representing the case where learned features are **invariant to spurious correlations**.
> - Other solutions can also minimize $\mathcal{L}_{\text{inv}}$ and achieve **high training accuracy by leveraging spurious correlations in the training set.**
> - By leveraging an **OOD validation set**, we can **select the model** corresponding to the stable solution $\hat{s}^{(i)} = \frac{1}{2}$, **ensuring the learning of invariant features.**
> - The dataset characteristics poses implicit constraint on $\hat{s}^{(i)}$ to **facilitate the model to converge to the stable solution.**
> - The "strange" results, "_optimizing a loss function about classification to reach the optimal, the corresponding prediction probability of the correct class is equal to the probability of random classification_", implies the encoder **learns invariant features by encoding $G_c$, and discards the spurious correlations**, which is a rational explanation under the **OOD setting.**
>
> ## **Why Eq. 64 takes expectation operation**
>
> In Eq. 63, the equation applies to a single sample (which is a random variable). In Eq. 64, we take the expectation over the training dataset to consider all samples.
>
> ## **Why Eq. 67 can be translated to 68**
>
> We agree with the reviewer that Eq. 67 cannot directly lead to Eq. 68. As discussed earlier, Eq. 68 represents just **one feasible solution to Eq. 67**, and the use of the "$\rightarrow$" symbol in Eq. 67 was inappropriate.
>
> **We have corrected this issue in the draft by removing the " $\rightarrow$ " symbol and adding discussions in line 1537 to clarify that $\sigma(\widehat{\mathbf{s}})_i = \frac{1}{K}$ is one feasible solution**.
>
> ---
>
> We hope this response resolves the reviewer’s concerns. **We will add the contents related in the above discussion on why we can select the stable solution to learn invariant features using $\mathcal{L}\_{inv}$ in our revised draft**. Please feel free to reach out if any further concerns. Thank you!

---

> > ### Comment · Reviewer_vWa5 · 2024-11-22
> > **Response**
> >
> > Thanks for your reply!  Although I still have some confusion, I don't think it will affect my decision to raise my score. I will raise my score to 6! Good Luck!

---

> > > ### Author Response · Authors · 2024-11-22
> > > **Response to Reviewer vWa5**
> > >
> > > Thank you for your valuable feedback and for the engaging discussions, which have given us the opportunity to further improve the quality of our work. We will incorporate the relevant points from our discussion into the revised draft. Thank you for your support and for raising the score!

---

> ### Author Response · Authors · 2024-11-26
> **Draft Revision**
>
> Dear Reviewer vWa5,
>
> Thank you once again for your valuable feedbacks and insightful comments. We have carefully revised and enhanced our draft to incorporate your valuable comments and the points in our discussion, including:
>
> - We have added more optimization details of the biased infomax in **Appendix H.2**.
> - We have added **Appendix F** to include comparisons with EQuAD.
> - We have added **Appendix E** to discuss the complexity of LIRS and provide a runtime analysis compared to various baseline methods.
> - We have updated **Appendix D.5** to reflect the discussions on the solutions of  $\mathcal{L}\_{inv}$.
>
>
> Please feel free to reach out if you have any further questions or suggestions. Thank you!

---

### Official Review · Reviewer_6k7Q · 2024-11-04

**Soundness:** 3
**Presentation:** 3
**Contribution:** 3
**Rating:** 6
**Confidence:** 2

**Summary:**

This paper presents LIRS (Learning graph Invariance by Removing Spurious features), a novel framework for improving Out-of-Distribution (OOD) generalization in graph representation learning. It critiques existing methods like IRM and VRex for capturing only limited invariant features. Instead of directly learning invariant features, LIRS first identifies and removes spurious features from those learned via Empirical Risk Minimization (ERM). The framework utilizes the biased infomax principle for learning spurious features and a class-conditioned cross-entropy loss for isolating invariant features. This work offers a robust approach to enhancing OOD generalization in graph learning.

**Strengths:**

1. The paper is well-written and easy to follow.
2. The strategy of decoupling invariant and spurious features is common in the field, but learning the spurious features first before removing them offers a new perspective. This approach provides a novel way to tackle the challenges of OOD generalization in graph representation learning.

**Weaknesses:**

Complexity: The process of first identifying spurious features and then removing them may introduce additional complexity in the learning pipeline. Discussing the computational efficiency and potential challenges in this two-stage process would be useful for understanding the trade-offs involved.

**Questions:**

Is there an analysis of how the multi-step process in this indirect learning approach affects the overall results? Additionally, during empirical training, what observations did you make? If the final performance does not improve, how would you analyze the sources of performance issues?

---

> ### Author Response · Authors · 2024-11-20
> **Rzesponse to Reviewer 6k7Q**
>
> *We sincerely thank you for your valuable feedback and constructive suggestions in reviewing our paper! Please see below for our responses to your comments and concerns.*
>
> ---
>
> >**W1: Efficiency and time cost of LIRS**
>
> **A1:** Thank you for your insightful comments.
>
> - The time complexity of LIRS depends on the specific GNN encoder employed. In this work, we use GIN, a 1-WL GNN encoder. The time complexity is **$\mathcal{O}(CkmF)$**, where $k$ is the number of GNN layers, $m$ is the average number of edges, and $F$ is the hidden feature dimension. LIRS introduces an additional constant factor $C > 1$ due to its 2-stage paradigm. Notably, **most other graph invariant learning algorithms also exhibit a complexity of $\mathcal{O}(CkmF)$**, as they require multiple GNN encoders for subgraph extraction and feature encoding respectively. **Therefore, the time cost gap between LIRS and other methods is not significant.**
> -  **We have added a detailed time analysis using the Motif-base and Ogbg-Molbbbp datasets, and revised our draft in Appendix E, highlighted in $\textcolor{blue}{\text{blue}}$.**
>
> **Table 1: Run time analysis on the Motif-base and Ogbg-Molbbbp datasets.**
> | Method | GOODMotif-base | OGBG-Molbbbp |
> |:---:|:---:|:---:|
> | ERM | 494.34±117.86 | 92.42±0.42 |
> | IRM | 968.94±164.09 | 151.84±7.53 |
> | Vrex | 819.94±124.54 | 129.13±12.93 |
> | GSAT | 1233±396.19 | 142.47±25.71 |
> | GREA | 1612.43±177.36 | 262.47±45.71 |
> | CIGA | 1729.14±355.62 | 352.14±93.32 |
> | AIA | 1422.34±69.33 | 217.36±11.04 |
> | EQuAD | 747.87±34.71 | 278.85±16.64 |
> | LIRS (Ours) | 504.87±24.04 | 421.32±19.86 |
>
> - On the Motif-base dataset, the biased infomax only requires 20 epochs training, making its time cost comparable to ERM.
> - On the Ogbg-Molbbbp dataset, the time cost of LIRS exceeds that of other methods since the biased infomax requires training for 100 epochs, and GSAT must be run to annotate node labels to enable adaptive biased infomax.
> - Below, **we provide a breakdown of the time cost for each stage:**
>
> **Table 2: Break down of running time for each stage in LIRS.**
> | Method | GOODMotif-base | OGBG-Molbbbp |
> |:---:|:---:|:---:|
> | Biased Infomax | 302.12 | 214.24 |
> | MinibatchKmeans+SVM | 6.63 | 3.04±0.35 |
> | Mark Nodes | - | 142.47±25.71 |
> | GNN Retraining | 196.12±24.04 | 61.57±6.86 |
>
> - The first three components account for the majority of the time cost; however, __they only need to be run once__. Retraining the GNN encoder is both **fast and stable**, with less variance as $\mathcal{L}\_{Inv}$ is merely a cross-entropy loss.
> - This presents a __key advantage of LIRS in terms of hyperparameter selection__. Specifically, most of OOD methods require hyperparameter search, and for most methods, **the process must be restarted entirely for each run, incurring significant time costs.** In contrast, for LIRS, **only the final GNN retraining step needs to be run multiple times for hyperparameter search, leading to a significantly reduced time cost when conducting multiple runs compared to other methods.**
>
> >**Q1: Some empirical observations and analysis during training**
>
> **A2:** Thank you for this thoughtful question. During model training, we have observed several noteworthy observations:
>
> - **Convergence rate**. Biased infomax tends to converge faster than standard infomax on synthetic datasets, while on real-world datasets, both require a similar number of epochs to learn spurious embeddings. This difference is likely due to the simplicity of the synthetic datasets.
> - **Impact of $\gamma$ in GCE:** The coefficient $\gamma$ in the GCE function plays a critical role, as it controls the reweighting strength for minority group samples.
> - To analyze performance issues, we primarily utilize synthetic datasets, which share similar characteristics with real-world datasets. Since we know the ground-truth environments in synthetic datasets, they allow us to quickly identify potential issues. For example:
>     - we can evaluate whether embeddings learned by biased infomax across different epochs capture spurious patterns by calculating the normalized mutual information (NMI) between clustering labels and environment labels to study the imprtance of training epochs.
>     - We can also adjust $\gamma$ for reweighting samples, given that we know what range of $\gamma$ is suitable, we can identify the importance of $\gamma$ by setting it to different values.
>
> - By validating our theoretical insights against experimental results on synthetic datasets, **we can more effectively identify critical components in LIRS, and improving its performance on real-world datasets.**
>
> ---
>
> We sincerely thank you for your careful review and your positive evaluation. We hope we have addressed your concerns on the efficiency point of our approach.

---

> > ### Author Response · Authors · 2024-11-25
> >
> > Dear Reviewer 6k7Q,
> >
> > As the discussion deadline approaches, we are wondering whether our responses have properly addressed your concerns regarding the efficiency of our work? Your feedback would be extremely helpful to us. If you have further comments or questions, we hope for the opportunity to respond to them.
> >
> > Many thanks,
> >
> > 7198 Authors

---

> > ### Comment · Reviewer_6k7Q · 2024-11-25
> >
> > Thank you for the response. My concern has been addressed, and I stand by my score.

---

> > > ### Author Response · Authors · 2024-11-25
> > >
> > > Dear Reviewer 6k7Q, thank you for your response and your valuable comments. We are pleased to hear that we were able to address your concern.

---

> > > > ### Author Response · Authors · 2024-11-26
> > > > **Draft Revision**
> > > >
> > > > Dear Reviewer 6k7Q,
> > > >
> > > > Thank you once again for your valuable feedbacks and insightful comments. We have carefully revised and enhanced our draft to incorporate your valuable comments and the points in our discussion, including:
> > > >
> > > > - We have added **Appendix E** to discuss the complexity of LIRS.
> > > > - We also provide a runtime analysis compared to various baseline methods in **Appendix E**.
> > > >
> > > > Please feel free to reach out if you have any further questions or suggestions. Thank you!

---

### Official Review · Reviewer_Nv5f · 2024-11-06

**Soundness:** 2
**Presentation:** 3
**Contribution:** 2
**Rating:** 6
**Confidence:** 3

**Summary:**

This paper proposes LIRS, a learning framework designed to Learn graph invariance, which consists of: a) The biased infomax principle, and b) The class-conditioned crossentropy loss, which elicit effective spuriosity learning and invariant learning respectively, aiming to learn more invariant features. Different from most existing approaches that directly learn the invariant features, LIRS takes an indirect approach by first learning the spurious features and then removing them from the ERM-learned features.

**Strengths:**

1.  The structure of the paper is clear and easy to follow.
2.  The paper conducts comprehensive experiments to demonstrate the performance of proposed method.

**Weaknesses:**

1.  The novelty seems limited. The core idea of LIRS is three parts, i.e., graph Invariance learning by removing spurious features, the biased infomax principle, and the class-conditioned cross-entropy loss. However, the first part is inspired by [1], the second part seems to be a combination of SCL[2] and DGI[3], the third part is the generalized cross-entropy (GCE) method[4]. The technical contribution is a little weak. It is recommended that the author provide a clearer description of the contribution.
2.  The proposed LIRS requires training the GNN encoder twice, which is more cost than the end-to-end methods. It is recommended that the authors provide an overall time cost.
3.  LIRS takes an indirect approach by first learning the spurious features by self-supervised and then removing them from the ERM-learned features. OOD-GCL[5] proposes a contrastive learning module with invariance regularization so that spurious correlations between latent factors in graph representation can be eliminated and OOD predictions in downstream tasks can be achieved. It is suggested that the authors analyze and compare OOD-GCL that directly learns invariant representations by self-supervised.

[1] Empowering graph invariance learning with deep spurious infomax. ICML, 2024.

[2] Supervised contrastive learning. NeurIPS, 2020.

[3] Deep graph infomax. ICLR, 2019.

[4] Generalized cross entropy loss for training deep neural networks with noisy labels. NeurIPS, 2018.

[5] Disentangled Graph Self-supervised Learning for Out-of-Distribution Generalization. ICML 2024.

**Questions:**

See the Weaknesses

---

> ### Author Response · Authors · 2024-11-20
> **Response to Reviewer Nv5f (Part 1/3)**
>
> *We sincerely thank you for your valuable feedback and constructive suggestions in reviewing our paper! Please see below for our responses to your comments and concerns.*
>
> ---
>
> >**W1: Novelty of LIRS**
>
>
> **A1:** Thank you for your thoughtful questions. Please see our response below.
>
> **Compare with EquAD**
>
> While LIRS and EQuAD adopt the same learning paradigm, **LIRS introduces significant technical contributions to this paradigm and address different research questions**, the unique contributions of LIRS are detailed as following:
>
> - **Motivation and new insights:** While EQuAD primarily addresses the limitations of existing OOD methods that are sensitive to varying spurious correlation strengths, it does not fully explain why the proposed learning paradigm is effective when spurious correlation strength remains stable. In contrast, **our work answers this important question and demonstrates that this learning paradigm enables learning a broader set of invariant features.**
> - **New Dataset Curation:** We curate the SPMotif-binary dataset based on the SPMotif datasets, which can **serve as a benchmark for future studies to evaluate the effectiveness of methods in learning a broader set of invariant features.**
> - **New Algorithm:** While EQuAD uses a standard infomax objective to learn spurious features, we propose a new algorithm that addresses the limitations of the vanilla infomax approach. Specifically, we introduce _the biased infomax_ to **overcome size constraints and further incorporate additional procedure to annotate critical nodes with adaptive thresholding to realize biased infomax in real-world datasets**.
> - **New Disentanglement Loss:** We **identify the limitations** of the cross-entropy loss in disentangling spurious features and propose a novel loss objective that is **more effective in learning invariant features**. Additionally, our proposed loss **does not compromise computational efficiency.**
>
> These differences highlight the unique contributions of our work compared to EQuAD.
>
>
> **Compare and contrast with SCL and DGI**
>
> We wish to clarify the conceptual and technical differences between our method and SCL/DGI as follows:
>
> - **Conceptual Differences:** SCL and DGI primarily **target in-distribution (ID) data** and are designed to improve performance on predictive tasks. In contrast, our proposed biased infomax is specifically **designed for OOD data**, with the goal of **learning environment-related features rather than features that directly benefit classification tasks**. This makes our approach conceptually distinct from SCL and DGI.
>
> - **Technical Differences:** While we agree with the reviewer that biased infomax might appear to be a combination of SCL and DGI, __it goes beyond a simple integration of these methods__. Specifically:
>    - In SCL, the contrastive loss operates at the inter-sample level, where explicit labels are available for each image. However, in our graph-level OOD setting, biased infomax operates at the node level, **where such labels are unavailable.**
>    - Due to the absence of labels, **we approximate critical nodes using GNN explainer with adaptive thresholding.** This additional procedure also distinct biased infomax from SCL and DGI.
>    - **The optimization objective of biased infomax** also differs from SCL and DGI as we need to maximize the MI between $\hat{G_s}$ and $\hat{h}_G$ while minimizing MI between $\hat{G_c}$ and $\hat{h}_G$.
>
> In summary, while biased infomax draws inspiration from SCL and DGI, **its conceptual motivation, technical design, and optimization goals differentiate it from SCL and DGI**.
>
> **The adoption of GCE loss**
>
> While we also utilize a generalized cross-entropy (GCE) loss with the same form as $\mathcal{L}_q$ in [1], the primary purpose in our work is to reweight data samples. We wish to clarify that **one of our main contributions lies in proposing the intra-class cross-entropy loss for disentangling spurious features**. **Any function capable of upweighting minority group samples could suffice for this purpose**, and we chose to adopt the GCE loss as an alternative, however we respectively claim that it does not undermine the novelty of our work.

---

> ### Author Response · Authors · 2024-11-20
> **Response to Reviewer Nv5f (Part 2/3)**
>
> > **W2: Time cost and efficiency of LIRS**
>
> **A4:** Thank you for raising this point.
>
> - The time complexity of LIRS depends on the specific GNN encoder employed. In this work, we use GIN, a 1-WL GNN encoder. The time complexity is **$\mathcal{O}(CkmF)$**, where $k$ is the number of GNN layers, $m$ is the average number of edges, and $F$ is the hidden feature dimension. LIRS introduces an additional constant factor $C > 1$ due to its 2-stage paradigm. Notably, **most other graph invariant learning algorithms also exhibit a complexity of $\mathcal{O}(CkmF)$**, as they require multiple GNN encoders for subgraph extraction and feature encoding respectively. **Therefore, the time cost gap between LIRS and other methods is not significant.**
> -  **We have added a detailed complexity analysis and running time analysis using the Motif-base and Ogbg-Molbbbp datasets, and revised our draft in Appendix E, highlighted in $\textcolor{blue}{\text{blue}}$.**
>
> **Table 1: Run time analysis on the Motif-base and Ogbg-Molbbbp datasets.**
> | Method | GOODMotif-base | OGBG-Molbbbp |
> |:---:|:---:|:---:|
> | ERM | 494.34±117.86 | 92.42±0.42 |
> | IRM | 968.94±164.09 | 151.84±7.53 |
> | Vrex | 819.94±124.54 | 129.13±12.93 |
> | GSAT | 1233±396.19 | 142.47±25.71 |
> | GREA | 1612.43±177.36 | 262.47±45.71 |
> | CIGA | 1729.14±355.62 | 352.14±93.32 |
> | AIA | 1422.34±69.33 | 217.36±11.04 |
> | OOD-GCL | 10813.14±28.12 | 8455.51±68.61 |
> | EQuAD | 747.87±34.71 | 278.85±16.64 |
> | LIRS (Ours) | 504.87±24.04 | 421.32±19.86 |
>
> - On the Motif-base dataset, the biased infomax only requires 20 epochs training, making its time cost comparable to ERM.
> - On the Ogbg-Molbbbp dataset, the time cost of LIRS exceeds that of other methods since the biased infomax requires training for 100 epochs, and GSAT must be run to annotate node labels to enable adaptive biased infomax. Below, **we provide a breakdown of the time cost for each stage:**
>
> **Table 2: Break down of running time for each stage in LIRS.**
> | Method | GOODMotif-base | OGBG-Molbbbp |
> |:---:|:---:|:---:|
> | Biased Infomax | 302.12 | 214.24 |
> | MinibatchKmeans+SVM | 6.63 | 3.04±0.35 |
> | Mark Nodes | - | 142.47±25.71 |
> | GNN Retraining | 196.12±24.04 | 61.57±6.86 |
>
> - The first three components account for the majority of the time cost; however, __they only need to be run once__. Retraining the GNN encoder is both **fast and stable**, with less variance as $\mathcal{L}\_{Inv}$ is merely a cross-entropy loss.
> - This presents a __key advantage of LIRS in terms of hyperparameter selection__. Specifically, most of OOD methods require hyperparameter search, and for most methods, **the process must be restarted entirely for each run, incurring significant time costs.** In contrast, for LIRS, **only the final GNN retraining step needs to be run multiple times for hyperparameter search, leading to a significantly reduced time cost when conducting multiple runs compared to other methods.**
>
> > **W3: Comparison with OOD-GCL**
>
> **A3:** Thank you for bringing up this related work.
>
>
> Despite OOD-GCL and LIRS both leveraging contrastive learning to learn invariant features, there are several key differences between the two methods:
>
> - **Problem Setting**. While LIRS uses contrastive learning (biased infomax) to identify spurious features, it relies on labeled data in the subsequent stage to effectively learn invariant features. In contrast, **OOD-GCL aims to learn invariant features without labeled data**, followed by  fine-tuning a linear classifier on downstream tasks. This distinction highlights different assumptions and goals in the design of the two methods.
> - **Effectiveness analysis.** OOD-GCL uses multiple channels and performs contrastive learning with invariance regularization to learn invariant features. However, **learning invariant representations entirely without labeled data is inherently challenging**, even with invariance regularization to minimize variance across channels and clusters. On the other hand, **LIRS leverages both spurious features from biased infomax and labeled data, enabling more effective learning of invariant features.**
>     - To evaluate the performance of OOD-GCL, **we have added experiments on 8 datasets used in our work, and updated our draft in Section 5, highlighted in $\textcolor{blue}{\text{blue}}$.** As there is no official code for OOD-GCL, we re-implemented it following the description in [2], using the same GNN architecture and hidden dimension size as LIRS. We tuned $K$ over $\\{2, 3\\}$, as LIRS employs up to 3 GNN encoders, while keeping the remaining hyperparameters consistent with [2].
>     - The results in the Table 3 below show that LIRS outperforms OOD-GCL across all datasets, **likely due to that LIRS has access to labeled data during training, while OOD-GCL uses labeled data only to fine-tune the linear layer**.

---

> ### Author Response · Authors · 2024-11-20
> **Response to Reviewer Nv5f (Part 3/3)**
>
> **Table 3: OOD performance on synthetic and real-world datasets.**
> | Method | Motif-basis | Motif-size | HIV-sca | HIV-size | Molbace-sca | Molbace-size | Molbbbp-sca | Molbbbp-size |
> |:---:|:---:|:---:|:---:|:---:|:---:|:---:|:---:|:---:|
> | OOD-GCL | 56.46±4.61 | 60.23±8.49 | 70.85±2.07 | 58.48±2.94 | 75.96±2.21 | 85.34±1.77 | 67.28±3.09 | 78.11±3.32 |
> | LIRS (Ours) | **75.51±2.19** | **74.95±7.69** | **72.82±1.61** | **66.64±1.44** | **81.91±1.98** | **88.77±1.64** | **71.04±0.76** | **82.19±1.57** |
>
> - **Efficiency analysis.** **We also evaluated the run time of OOD-GCL, with results presented in Table 1**. As can be seen, **OOD-GCL is significantly slower than other methods** because it requires multiple rounds of clustering in every epoch and additional invariance regularization, which must be applied for each channel and cluster. In contrast, **LIRS only performs preprocessing steps once, and the GNN retraining phase is efficient, greatly facilitating hyperparameter search.**
>
> ---
>
> We sincerely thank you for your careful review and insightful comments.  We hope we have addressed your concerns on the novelty and efficiency of our approach.
>
> ---
>
> __References__
>
> [1] Zhang et al., Generalized cross entropy loss for training deep neural networks with noisy labels. NeurIPS, 2018.
>
> [2] Li, et al., Disentangled Graph Self-supervised Learning for Out-of-Distribution Generalization. ICML2024.

---

> > ### Author Response · Authors · 2024-11-25
> >
> > Dear Reviewer Nv5f,
> >
> > As the discussion deadline approaches, we are wondering whether our responses have properly addressed your concerns regarding the efficiency and novelty of our work? Your feedback would be extremely helpful to us. If you have further comments or questions, we hope for the opportunity to respond to them.
> >
> > Many thanks,
> >
> > 7198 Authors

---

> > ### Comment · Reviewer_Nv5f · 2024-11-25
> > **Thanks for the authors' responses**
> >
> > Thanks for the authors' responses which addressed most of my concerns, I will boost my score to 6. I hope our discussions can be included in the revised version.

---

> > > ### Author Response · Authors · 2024-11-25
> > >
> > > Dear Reviewer Nv5f,
> > >
> > > Thank you for your valuable feedback and insightful comments, which have greatly helped us improve the quality of our paper and enrich its content. We will surely incorporate **all the relevant points** from our discussion into the revision. We sincerely appreciate your support and for raising your score!

---

> ### Author Response · Authors · 2024-11-26
> **Draft Revision**
>
> Dear Reviewer Nv5f,
>
> Thank you once again for your valuable feedbacks and insightful comments. We have carefully revised and enhanced our draft to incorporate your valuable comments and the points in our discussion, including:
>
> - We have added **Appendix F** to include comparisons with EQuAD, SCL, DGI, and OOD-GCL.
> - We have added **Appendix E** to discuss the complexity of LIRS and provide a runtime analysis compared to various baseline methods.
> - We have included the results of OOD-GCL in the main results (in **Table 1**) and added a performance analysis of OOD-GCL in **Section 5.2**.
>
> Please feel free to reach out if you have any further questions or suggestions. Thank you!

---

### Official Review · Reviewer_8xLE · 2024-11-19

**Soundness:** 3
**Presentation:** 3
**Contribution:** 3
**Rating:** 6
**Confidence:** 3

**Summary:**

This paper introduces LIRS, a framework designed to improve Out-of-Distribution (OOD) generalization in graph representation learning by decoupling spurious and invariant features. The authors propose a two-step approach: (1) identifying and removing spurious features using a biased infomax principle, and (2) isolating invariant features through a class-conditioned cross-entropy loss. This method is positioned as an alternative to learning invariant features directly from graph representations, offering a new perspective on handling spurious correlations in graph learning.

**Strengths:**

- The paper is well-structured and easy to follow, with clear motivation for the proposed approach.

- The strategy of first learning and removing spurious features is an interesting and novel perspective on OOD generalization in graph learning. This decoupling of spurious and invariant features is a unique contribution to the field, offering a potential advantage over direct approaches.

**Weaknesses:**

- The two-stage process introduces additional complexity in the learning pipeline, which may increase computational costs and practical challenges. A more detailed discussion of potential trade-offs, such as time complexity and training efficiency, would be beneficial.

- The novelty of the contribution could be clearer, as parts of the method bear resemblance to existing work, particularly in the use of biased infomax and cross-entropy loss. A deeper comparison with related approaches, such as OOD-GCL or EQuAD, would strengthen the paper.

**Questions:**

- How does the multi-step process of identifying and removing spurious features impact overall model efficiency? Have any time or resource costs been benchmarked compared to existing methods?

- Can the authors provide a more detailed explanation of how LIRS differs from related works like EQuAD or OOD-GCL?

---

> ### Author Response · Authors · 2024-11-20
> **Response to Reviewer 8xLE (Part 1/2)**
>
> *We sincerely thank you for your valuable feedback and constructive suggestions in reviewing our paper! Please see below for our responses to your comments and concerns.*
>
> ---
>
> >**W1&Q1: Efficiency of LIRS**
>
> **A1:** Thank you for your thoughtful questions.
>
> - The time complexity of LIRS depends on the specific GNN encoder employed. In this work, we use GIN, a 1-WL GNN encoder. The time complexity is **$\mathcal{O}(CkmF)$**, where $k$ is the number of GNN layers, $m$ is the average number of edges, and $F$ is the hidden feature dimension. LIRS introduces an additional constant factor $C > 1$ due to its 2-stage paradigm. Notably, **most other graph invariant learning algorithms also exhibit a complexity of $\mathcal{O}(CkmF)$**, as they require multiple GNN encoders for subgraph extraction and feature encoding respectively. **Therefore, the time cost gap between LIRS and other methods is not significant.**
> -  **We have added a detailed time analysis using the Motif-base and Ogbg-Molbbbp datasets.**
>
> **Table 1: Run time analysis on the Motif-base and Ogbg-Molbbbp datasets.**
> | Method | GOODMotif-base | OGBG-Molbbbp |
> |:---:|:---:|:---:|
> | ERM | 494.34±117.86 | 92.42±0.42 |
> | IRM | 968.94±164.09 | 151.84±7.53 |
> | Vrex | 819.94±124.54 | 129.13±12.93 |
> | GSAT | 1233±396.19 | 142.47±25.71 |
> | GREA | 1612.43±177.36 | 262.47±45.71 |
> | CIGA | 1729.14±355.62 | 352.14±93.32 |
> | AIA | 1422.34±69.33 | 217.36±11.04 |
> | OOD-GCL | 10813.14±28.12 | 8455.51±68.61 |
> | EQuAD | 747.87±34.71 | 278.85±16.64 |
> | LIRS (Ours) | 504.87±24.04 | 421.32±19.86 |
>
>
> - On the Motif-base dataset, the biased infomax only requires 20 epochs training, making its time cost comparable to ERM.
> - On the Ogbg-Molbbbp dataset, the time cost of LIRS exceeds that of other methods since the biased infomax requires training for 100 epochs, and GSAT must be run to annotate node labels to enable adaptive biased infomax.
> - Below, **we provide a breakdown of the time cost for each stage:**
>
> **Table 2: Break down of running time for each stage in LIRS.**
> | Method | GOODMotif-base | OGBG-Molbbbp |
> |:---:|:---:|:---:|
> | Biased Infomax | 302.12 | 214.24 |
> | MinibatchKmeans+SVM | 6.63 | 3.04±0.35 |
> | Mark Nodes | - | 142.47±25.71 |
> | GNN Retraining | 196.12±24.04 | 61.57±6.86 |
>
> - The first three components account for the majority of the time cost; however, __they only need to be run once__. Retraining the GNN encoder is both **fast and stable**, with less variance as $\mathcal{L}_{Inv}$ is merely a cross-entropy loss.
> - This presents a key advantage of LIRS __in terms of hyperparameter selection__. Specifically, most of OOD methods require hyperparameter search, and for most methods, **the process must be restarted entirely for each run, incurring significant time costs.** In contrast, for LIRS, **only the final GNN retraining step needs to be run multiple times for hyperparameter search, leading to a significantly reduced time cost when conducting multiple runs compared to other methods.**
>
> **We have added complexity analysis with new experimental results on running time of various methods in Appendix E, highlighted in $\textcolor{blue}{\text{blue}}$.**

---

> > ### Author Response · Authors · 2024-11-25
> >
> > Dear Reviewer 8xLE,
> >
> > As the discussion deadline approaches, we are wondering whether our responses have properly addressed your concerns regarding the efficiency and novelty of our work? Your feedback would be extremely helpful to us. If you have further comments or questions, we hope for the opportunity to respond to them.
> >
> > Many thanks,
> >
> > 7198 Authors

---

> ### Author Response · Authors · 2024-11-20
> **Response to Reviewer 8xLE (Part 2/2)**
>
> > **W2&Q2: Compare and contrast with EQuAD and OOD-GCL**
>
> **A2:** Thank you for this insightful question. Please see our response below.
>
> **Compare with EQuAD**
>
> While LIRS and EQuAD adopt the same learning paradigm, **LIRS introduces significant technical contributions to this paradigm and address different research questions**, the unique contributions of LIRS are detailed as following:
>
> - **Motivation and new insights:** While EQuAD primarily addresses the limitations of existing OOD methods that are sensitive to varying spurious correlation strengths, it does not fully explain why the proposed learning paradigm is effective when spurious correlation strength remains stable. In contrast, **our work answer this important question and demonstrates that this learning paradigm enables learning a broader set of invariant features.**
> - **New Dataset Curation:** We curate the SPMotif-binary dataset based on the SPMotif datasets, which can **serve as a benchmark for future studies to evaluate the effectiveness of methods in learning a broader set of invariant features.**
> - **New Algorithm:** While EQuAD uses a standard infomax objective to learn spurious features, we propose a new algorithm that addresses the limitations of the vanilla infomax approach. Specifically, we introduce _the biased infomax_ to **overcome size constraints and further incorporate additional procedure to annotate critical nodes with adaptive thresholding to realize biased infomax in real-world datasets**.
> - **New Disentanglement Loss:** We **identify the limitations** of the cross-entropy loss in disentangling spurious features and propose a novel loss objective that is **more effective in learning invariant features**. Additionally, our proposed loss **does not compromise computational efficiency.**
>
> These differences highlight the unique contributions of our work compared to EQuAD.
>
> **Compare with OOD-GCL**
>
> Although both OOD-GCL and LIRS leverage contrastive learning to learn invariant features, there are several key differences between these two methods:
>
> - **Problem Setting**. While LIRS uses contrastive learning (biased infomax) to identify spurious features, it relies on labeled data in the subsequent stage to effectively learn invariant features. In contrast, OOD-GCL aims to learn invariant features without labeled data, followed by  fine-tuning a linear classifier on downstream tasks. This distinction highlights different assumptions and goals in the design of the two methods.
> - **Effectiveness analysis.** OOD-GCL uses multiple channels and performs contrastive learning with invariance regularization to learn invariant features. However, **learning invariant representations entirely without labeled data is inherently challenging**, even with invariance regularization to minimize variance across channels and clusters. On the other hand, **LIRS leverages both spurious features from biased infomax and labeled data, enabling more effective learning of invariant features.**
>     - To evaluate the performance of OOD-GCL, **we have added experiments on 8 datasets used in our work, and updated the draft in Section 5, highlighted in $\textcolor{blue}{\text{blue}}$**. As there is no official code for OOD-GCL, we re-implemented it following the description in [1], using the same GNN architecture and hidden dimension size as LIRS. We tuned $K$ over $\\{2, 3\\}$, as LIRS employs up to 3 GNN encoders, while keeping the remaining hyperparameters consistent with [1].
>     - The results in the table below show that **LIRS outperforms OOD-GCL across all datasets, likely due to that LIRS is able to access to labeled data during training**, while OOD-GCL uses labeled data only to fine-tune the linear layer.
>
> **Table 3: Test performance of OOD-GCL**
> | Method | Motif-basis | Motif-size | HIV-sca | HIV-size | Molbace-sca | Molbace-size | Molbbbp-sca | Molbbbp-size |
> |:---:|:---:|:---:|:---:|:---:|:---:|:---:|:---:|:---:|
> | OOD-GCL | 56.46±4.61 | 60.23±8.49 | 70.85±2.07 | 58.48±2.94 | 75.96±2.21 | 85.34±1.77 | 67.28±3.09 | 78.11±3.32 |
> | LIRS (Ours) | **75.51±2.19** | **74.95±7.69** | **72.82±1.61** | **66.64±1.44** | **81.91±1.98** | **88.77±1.64** | **71.04±0.76** | **82.19±1.57** |
>
> - **Efficiency analysis. We also evaluated the run time of OOD-GCL, with results presented in Table 1**. As can be seen, **OOD-GCL is significantly slower than other methods** because it requires multiple rounds of clustering in every epoch and additional invariance regularization, which must be applied for each channel and cluster. In contrast, **LIRS only performs preprocessing steps once, and the GNN retraining phase is efficient, greatly facilitating hyperparameter search.**
>
> ---
>
> We sincerely thank you for your careful review and insightful comments.  We hope we have addressed your concerns on the novelty and efficiency of our approach.
>
> ___
>
> __References__
>
> [1] Li, et al., Disentangled Graph Self-supervised Learning for Out-of-Distribution Generalization. ICML2024.

---

> > ### Author Response · Authors · 2024-11-26
> > **Draft Revision**
> >
> > Dear Reviewer 8xLE,
> >
> > Thank you once again for your valuable feedbacks and insightful comments. We have carefully revised and enhanced our draft to incorporate your valuable comments and the points in our discussion, including:
> >
> >
> > - We have added **Appendix E** to discuss the complexity of LIRS and provide a runtime analysis compared to various baseline methods.
> > - We have added **Appendix F** to include comparisons with EQuAD and OOD-GCL.
> > - We have included the results of OOD-GCL in the main results (in **Table 1**) and added a performance analysis of OOD-GCL in **Section 5.2**.
> >
> > Please feel free to reach out if you have any further questions or suggestions. Thank you!

---

### Meta-Review · Area_Chair_LwN6 · 2024-12-26

**Metareview:**

The paper introduces LIRS, a novel framework for improving Out-of-Distribution (OOD) generalization in graph representation learning by decoupling spurious and invariant features. The overall soundness of the paper is generally good, with solid experimental results supporting the proposed framework. The method's novelty lies in its decoupling strategy, though some concerns about the method's originality and efficiency were raised. Reviewers pointed out that certain aspects, like the use of biased infomax and cross-entropy loss, are inspired by existing works, reducing the perceived novelty. Additionally, the proposed approach's complexity and computational costs were highlighted as potential drawbacks, especially considering the need for multiple training steps. Despite these weaknesses, there was a consensus that the paper has sufficient merit to be accepted.

**Additional Comments On Reviewer Discussion:**

The authors are encouraged to provide a more detailed comparison with related works such as OOD-GCL and EQuAD to better highlight the novelty and unique contributions of their approach. Additionally, a discussion on the computational efficiency of LIRS, particularly regarding the multi-step training process and potential trade-offs in time complexity, would be valuable. The paper should also clarify the implementation of the biased infomax principle and justify its selection over alternative methods. Further, the authors should explain how cross-entropy is used to isolate invariant features and address concerns raised about Proposition 7, particularly regarding the equivalence of minimizing cross-entropy and maximizing conditional entropy.

---

### Decision · Program_Chairs · 2025-01-22

Accept (Poster)